# Cell-selective proteomics segregates pancreatic cancer subtypes by extracellular proteins in tumors and circulation

Jonathan J. Swietlik[1,10], Stefanie Bärthel [2,3,4,10], Chiara Falcomatà [2,3,4,10], Diana Fink[5], Ankit Sinha[6], Jingyuan Cheng[1], Stefan Ebner[5], Peter Landgraf[7], Daniela C. Dieterich[7,8], Henrik Daub[9], Dieter Saur [2,3,4] ✉ & Felix Meissner [1,5] ✉

Cell-selective proteomics is a powerful emerging concept to study heterocellular processes in tissues. However, its high potential to identify non-cell-autonomous disease mechanisms and biomarkers has been hindered by low proteome coverage. Here, we address this limitation and devise a comprehensive azidonorleucine labeling, click chemistry enrichment, and mass spectrometry-based proteomics and secretomics strategy to dissect aberrant signals in pancreatic ductal adenocarcinoma (PDAC). Our in-depth co-culture and in vivo analyses cover more than 10,000 cancer cell-derived proteins and reveal systematic differences between molecular PDAC subtypes. Secreted proteins, such as chemokines and EMT-promoting matrisome proteins, associated with distinct macrophage polarization and tumor stromal composition, differentiate classical and mesenchymal PDAC. Intriguingly, more than 1,600 cancer cell-derived proteins including cytokines and pre-metastatic niche formation-associated factors in mouse serum reflect tumor activity in circulation. Our findings highlight how cell-selective proteomics can accelerate the discovery of diagnostic markers and therapeutic targets in cancer.

Cells in multicellular organisms adapt their phenotypes and function by crosstalk with other cell types. Short- and long-ranged intercellular signals are an integral part of organismal homeostasis and, when altered, drive the pathogenesis of diverse diseases. For example, in cancer, vivid interactions between transformed cells and non-transformed stromal cells promote or inhibit tumor development, metastasis, and the efficacy of drugs.

A rising incidence and high lethality make pancreatic ductal adenocarcinoma (PDAC) one of the leading causes of cancer-related deaths[1]. Since PDAC is typically discovered in advanced stages and refractory to most treatment modalities, there is a pressing need for more effective therapy and biomarkers that allow early detection. However, hallmark features of PDAC, such as a dense and fibrotic stroma, an immunosuppressive tumor microenvironment (TME), and often low neoplastic cellularity, exacerbate its molecular characterization and therapy development[2,3]. Based on the transcriptional profile and pathological features, PDAC is stratified into two major molecular subtypes[4]. Classical PDAC is characterized by a well-

[1]Experimental Systems Immunology, Max Planck Institute of Biochemistry, Martinsried, Germany. [2]Division of Translational Cancer Research, German Cancer Research Center and German Cancer Consortium, Heidelberg, Germany. [3]Chair of Translational Cancer Research and Institute of Experimental Cancer Therapy, University Hospital Rechts der Isar, School of Medicine, Technical University of Munich, Munich, Germany. [4]Center for Translational Cancer Research (TranslaTUM), School of Medicine, Technical University of Munich, Munich, Germany. [5]Institute of Innate Immunity, Department of Systems Immunology and Proteomics, Medical Faculty, University of Bonn, Bonn, Germany. [6]Department of Proteomics and Signal Transduction, Max Planck Institute of Biochemistry, Martinsried, Germany. [7]Institute for Pharmacology and Toxicology, Otto-von-Guericke-University Magdeburg, Magdeburg, Germany. [8]Center for Behavioral Brain Sciences, Magdeburg, Germany. [9]NEOsphere Biotechnologies GmbH, Martinsried, Germany. [10]These authors contributed equally: Jonathan J. Swietlik, Stefanie Bärthel, Chiara Falcomatà. ✉e-mail: dieter.saur@tum.de; felix.meissner@uni-bonn.de

differentiated histopathology and epithelial gene expression signature. In contrast, mesenchymal (basal-like) PDAC shows an undifferentiated, non-glandular histology, a mesenchymal gene expression profile, and is associated with a poor prognosis and high resistance to standard-of-care chemotherapy compared to the classical subtype[5–9]. Despite the substantial clinicopathological differences between the two PDAC subtypes, the underlying differences in the intercellular signaling of cancer cells with their TME have not been studied systematically so far.

Important insights into tumor cell composition and phenotype have been gained by systems-wide transcriptional approaches. However, the correlation between mRNA and protein copy numbers can vary widely[10,11], especially for proteins with roles in intercellular crosstalk[12,13]. Therefore, systems-wide and unbiased tools for

comprehensive quantitative protein analyses can provide unique perspectives on the context-dependent crosstalk of cancer cells with their microenvironment[14]. Mass spectrometry (MS)-based proteomics is today's gold standard for high throughput protein analysis and has significantly improved our understanding of cancer pathogenesis[15–18]. The combination of proteomics with cell-selective metabolic protein labeling strategies promises to resolve context-dependent cell behavior and interaction in complex heterocellular systems like tumors. One of the emerging methods uses the specially engineered methionyl-tRNA-synthetase[L274G] (MetRS*), which enables the time-controlled and cell-specific introduction of the non-canonical amino acid azidonorleucine (Anl) into proteomes[19–21]. Azide-alkyne click chemistry allows the subsequent extraction of MetRS*-expressing cell-derived proteins from cell mixtures. Successful application in living

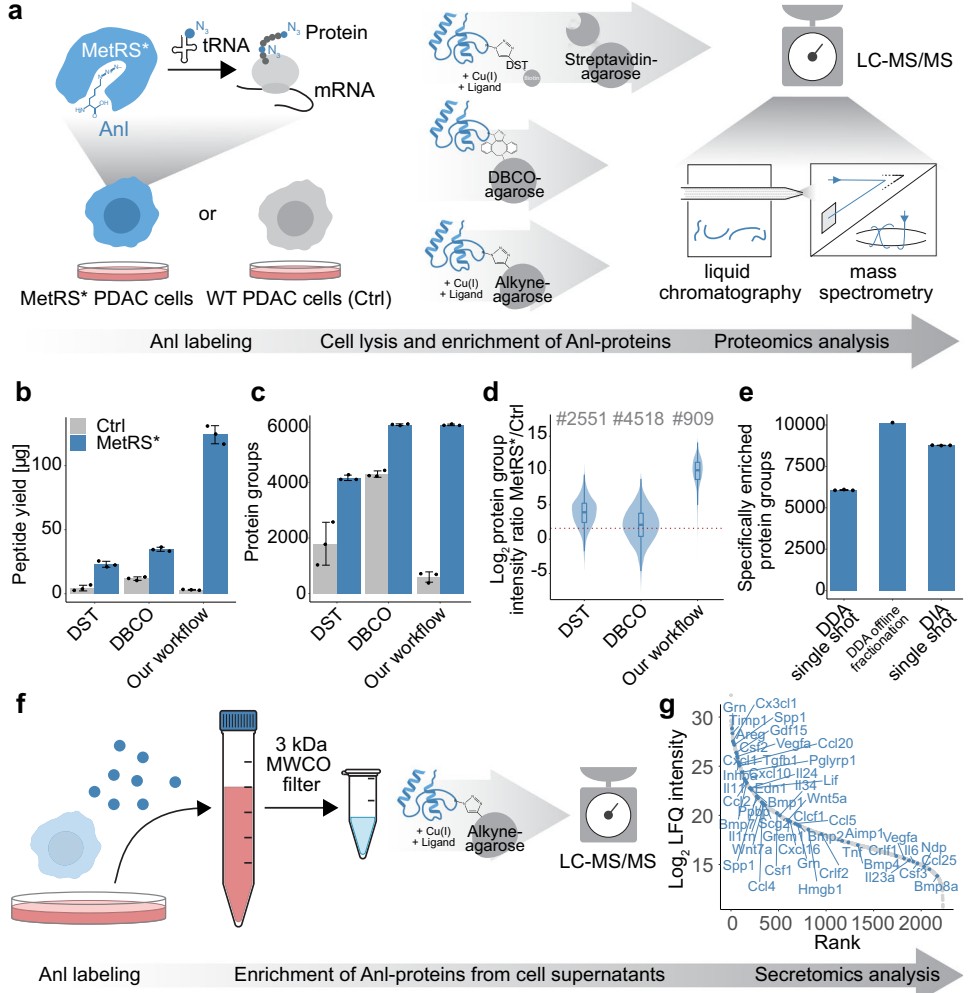

**Fig. 1 | A sensitive workflow for comprehensive cell type-selective proteomics and secretomics. a** Scheme of cell-selective proteomics workflows: The methionyl-tRNA synthetase[L274G] (MetRS*) activates azidonorleucine (Anl) by loading it onto methionyl-tRNAs. MetRS*-expressing cells incorporate Anl as a methionine substitute into newly synthesized proteins. Lentivirally transduced primary MetRS*-expressing or wild-type (Ctrl) PDAC cells isolated from mouse tumors with a conditional pancreatic expression of Kras[G12D] were grown for 8 h in Met-depleted medium supplemented with 4 mM Anl. $1 \times 10^7$ MetRS* and Ctrl cells were processed by DST enrichment, DBCO enrichment, and our improved alkyne-agarose CuAAC enrichment protocols ($n = 3$, workflow replicates). **b** Peptide yields (mean ± SD) determined by absorbance at 280 nm after enrichment, digestion, and solid phase peptide extraction. **c** Identified protein groups (mean ± SD) after MS-based analysis using 2 h chromatographic gradient length and data-dependent acquisition (DDA). **d** Intensity ratios of proteins identified in MetRS* and Ctrl samples. Counts of

overlapping identifications with ratios are indicated. **e** Specifically enriched protein groups (exclusive or >3-fold higher intensity compared to Ctrl samples) identified after alkyne-agarose enrichment and single run DDA, DDA analysis of 16 fractions separated by offline high-pH reverse phase chromatography, or single run data-independent acquisition (DIA) (mean ± SD, fractionation $n = 1$, single shots $n = 3$, workflow replicates). The latter was used for all further experiments. **f** Scheme of cell-selective secretomics workflow: MetRS* and Ctrl 8661 PDAC cells were cultured for 8 h in 5% FBS containing Met-depleted medium with 4 mM Anl ($n = 3$, workflow replicates). MetRS*-expressing cell-derived Anl-proteins were enriched from cell supernatants after buffer exchange and concentration. **g** Specifically-enriched PDAC cell-released proteins ranked by label-free quantification (LFQ) intensity. Proteins with cytokine function are indicated. Source data are provided as a Source Data file.

animals provided evidence for broad tissue compatibility[22] and revealed, for example, differential expression of 200 proteins in hippocampal excitatory neurons in mice exposed to an enriched environment[23]. In contrast to cell-sorting-based strategies such as FACS or MACS, intact tissues are snap-frozen directly after harvesting and subsequently lysed without cell dissociation. This effectively avoids cell-damage-related losses, selection bias for more robust cell populations, and potential protein expression or modification state artifacts by stresses and environmental changes during the enzymatic and mechanical treatment necessary to extract cells from tissues[24–27]. However, the achieved proteome coverage has generally been low, and even the deepest studies remained under 4000 specifically enriched proteins[23,28], leaving open the feasibility of comprehensive Anl enrichment-based proteomics analysis.

Here, we developed an improved workflow that enables an unprecedented proteomics depth for cell type-specific cellular proteome and secretome profiling in vitro and in vivo. This vastly increased the detection capacity of often low abundant intercellular signaling proteins such as secreted cytokines or receptors and therefore raised the potential for MetRS*/ Anl-based cellular communication analyses. We applied our approaches in the context of primary PDAC co-culture and orthotopic transplantation models and demonstrated unique advantages in capturing extracellular proteins compared to conventional cell sorting-based proteomics. We used the strength of our comprehensive cell-type specific proteomics workflow to reveal functional differences between classical and mesenchymal PDAC subtypes in tumors and circulation, such as context-specific secretion of cancer cell-derived EMT-promoting molecules and immunomodulators that correlated with differential immune cell recruitment in vivo, as well as distinct qualitative and quantitative contributions of cancer cell-derived proteins to the tumor extracellular matrix (ECM).

## Results

### An improved workflow enables highly efficient and specific cell-selective enrichment of proteins

Conceptually, methionyl-tRNA synthetase[L274G] (MetRS*)-based azidonorleucine (Anl) labeling offers unique possibilities for analyzing intercellular interactions in complex heterocellular systems. However, the achieved proteomic depth in our initial experiments and previously published MetRS*-based studies did not exceed 4000 proteins[23,28] and was therefore significantly lower than state-of-the-art with modern mass spectrometers and software[29], limiting the discovery potential. Hence, we set out to identify and overcome technical bottlenecks.

We first evaluated the Anl-incorporation rates of MetRS*-expressing cells in vitro by conventional MS-based shotgun proteomics without specific enrichment. Quantifying Anl-containing peptides compared to their unmodified counterparts showed that Anl incorporation was indeed highly specific to MetRS*-expressing cells but much slower than the incorporation of methionine (L-methionine-methyl-$^{13}$C,d$_3$) or the MetRS*-independent Met-substitute azidohomoalanine (Aha) (Supplementary Fig. 1a, b). Furthermore, Anl labeling is strongly dosage-dependent and reduced with methionine competition, as shown in previous studies[19,30,31]. We reasoned that the Anl-protein abundance would be very low in most applications, especially in vivo, considering often pronounced cell type heterogeneity and limited Anl bioavailability in tissues. Consequently, the demands for both recovery and specificity of the enrichment workflow are very high when aiming for deep proteomics analyses. We chose a straightforward copper(I)-catalyzed azide-alkyne cycloaddition (CuAAC) and alkyne agarose-based strategy for scalability and high reaction rates[32] as the basis for protocol optimization. We individually evaluated key experimental steps to improve protein extraction from tissue and click chemistry efficiency by systematic implementation of previous findings[33,34] and empirical testing of

reactant ratios, buffer components, and new reagents, including next-generation Cu(I)-stabilizing agents[35].

A direct comparison of our improved alkyne-agarose CuAAC protocol with frequently used dibenzocyclooctyne (DBCO) resin- and cleavable disulfide biotin alkyne-tag (DST)-based procedures (Fig. 1a) demonstrated substantial advantages: Using MetRS*-expressing and negative control wild-type primary PDAC cells that were both incubated in Anl-containing media, our protocol showed minimal unspecific background and a drastically increased yield of specifically enriched peptides (Fig. 1b). This advantage translated well into the MS analysis: While DST-based enrichment provided good specificity but reduced overall coverage, DBCO-based enrichment led to many identifications in both MetRS* and negative control samples, concordant with higher side reactivity of strained alkynes[32,36]. In contrast, our protocol yielded deep proteome coverage but with the fewest identifications in negative controls (Fig. 1c). The technical reproducibility was equal to or better than alternative protocols, with 84% of all MetRS* sample identifications quantified in all three replicates and a median precursor coefficient of variation (CV) of 11.5% (Supplementary Fig. 2a, b). Importantly, low overlap and high-intensity differences of proteins identified in both MetRS* samples and controls demonstrated very low background interference from unspecific enrichment with our workflow (Fig. 1d). We defined proteins as specifically enriched if they were exclusively identified in MetRS* samples or quantified with at least threefold higher intensity than in negative controls and excluded all other proteins from further analysis, as described by Alvarez-Castelao et al.[23,37]. Accordingly, our workflow identified a total of 6576 specifically enriched protein groups (compared to 4416 and 4736 with DST- or DBCO-based enrichment, respectively), including almost all proteins covered with both other methods together plus 1039 exclusive identifications (Supplementary Fig. 3).

To optimize deep proteomics investigations, we combined our workflow with offline high pH reverse phase fractionation of peptides after enrichment and digestion, resulting in the identification of 10,146 specifically enriched protein groups, demonstrating exceptional proteome coverage (Fig. 1e). Furthermore, using a data-independent acquisition (DIA) method, we achieved an average of 8770 specifically enriched protein groups per sample in 2-h runs without fractionation. The use of DIA also improved data completeness between replicates and decreased precursor CVs compared to data-dependent acquisition (DDA) (Supplementary Fig. 2c, d).

To further evaluate the technical reproducibility of our enrichment, we repeated the experiment with ten negative control replicates, confirming the previously observed high signal-to-noise ratio between specifically enriched proteins and unspecific background (Supplementary Fig. 4a, c). While the very low signal intensity in negative controls caused more stochastic identifications than in MetRS* samples (Supplementary Fig. 4b), results remained very consistent when control samples were divided into groups of three and used separately to evaluate background interference in MetRS* samples (Supplementary Fig. 4c). The vast majority of proteins with sparse identifications in controls had high ratios far above our chosen specificity cutoff (Supplementary Fig. 4d). Conversely, the majority of proteins with lower MetRS*/Ctrl ratios had very high data completeness. Thus, not only were there a very limited number of proteins with higher background interference overall, but the controls were also effective in capturing most of these proteins consistently.

We applied the described filtering strategy to all subsequent MetRS* experiments in this study, using at least three experiments with wild-type cells as negative controls for corresponding MetRS* sample groups to define specifically enriched proteins and ensure high confidence in cell selectivity. To enable both very deep and MS time-efficient analysis with high throughput for larger-scale experiments, we used our enrichment workflow together with DIA single-shot analyses.

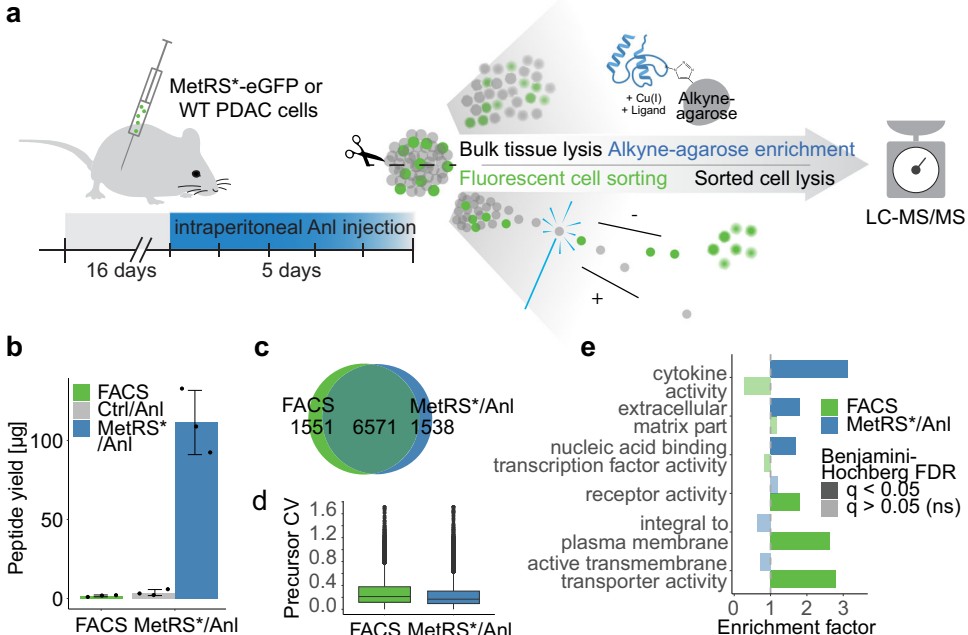

**Fig. 2 | MetRS\*- and FACS-based cancer cell-selective proteomics in vivo.**
**a** Scheme of PDAC transplantation for cell-selective proteomics: MetRS\*-eGFP
expressing 8661 PDAC (>90% eGFP-positive cells before transplantation, see Sup-
plementary Fig. 5a) and wild-type (Ctrl) cells were orthotopically transplanted into
fully immunocompetent syngeneic mice (*n* = 3, biological replicates). After a
16 days tumor growth period, mice were interperitoneally injected with Anl twice
daily for 5 days. Afterward, tumors were harvested and cut in half. One half was
snap-frozen for subsequent click chemistry enrichment, the other half was used

fresh for cell dissociation and eGFP-FACS. **b** Peptide yields (mean ± SD) determined
by absorbance at 280 nm. **c** Exclusively identified and overlap of (specifically
enriched) cancer cell-derived protein groups with either method. **d** Distribution of
precursor coefficients of variation (CVs) between biological replicates. **e** Analysis of
enriched GO annotations (Fisher's exact test) within exclusively identified proteins
with either method compared to all other identified proteins (full list in Supple-
mentary Data 4). Source data are provided as a Source Data file.

## Comprehensive cell-selective secretomics analysis in serum-containing media

Encouraged by the strongly increased specific peptide recovery, we next
aimed to adapt our methods for investigations of intercellular signaling,
specifically for the comprehensive analysis of secreted proteins. Pre-
viously, non-cell-selective incorporation of azide amino acids has
improved the detection of comparably low abundant secreted cellular
proteins in the presence of highly abundant serum proteins in serum-
containing conditioned media[38]. MetRS\*-based Anl-labeling could
expand this concept for cell-selective analyses in heterocellular systems
such as co-culture experiments. To establish proof-of-concept for in-
depth secretomics with our enrichment protocol, we analyzed super-
natants of primary PDAC cells in the presence of 5% serum (Fig. 1f). This
yielded deep coverage of PDAC cell-released proteins, with a total of
2229 specifically enriched protein groups and 788 protein groups
annotated with UniProtKB Keywords "secreted" and/or "signal." Of
those, 103 protein groups are known ligands for intercellular commu-
nication according to CellPhoneDB[39], including 46 with described
cytokine function (Fig. 1g). Despite their often small size and low
abundance, 83 (81%) and 41 (89%) of the detected intercellular signaling
proteins and cytokines were identified with at least two peptides.

## Increased yields and extracellular protein coverage of MetRS\*-based cell-selective proteomics compared to FACS in vivo

A key feature of MetRS\*-based Anl labeling is its applicability in living
animals. As shown previously in Falcomatà and Bärthel et al.[40], we
modeled molecular PDAC subtypes in vivo by orthotopic transplan-
tation of primary low-passaged cancer cells in the pancreas of fully
immunocompetent syngeneic mice. We evaluated our enrichment
workflow with tissue samples from this model by directly comparing
Anl-based enrichments with conventional fluorescence-activated cell
sorting (FACS) from MetRS\* and eGFP co-expressing cells. After cell

injection and an initial tumor growth period, we supplemented Anl by
intraperitoneal injection and then used one-half of each tumor for Anl-
enrichment or FACS (Fig. 2a). In total, 13–17% of the dissociated cells
were cancer cells, as indicated by eGFP-fluorescence (Supplemen-
tary Fig. 5).

Peptide yields revealed striking differences with an over 50-fold
higher average recovery of cancer cell-derived proteins by click
chemistry enrichment compared to FACS, indicating significant cell
losses during the dissociation and sorting procedure (Fig. 2b). How-
ever, both methods yielded a sufficient peptide amount for single-shot
proteomics analyses with modern MS instrumentation. Both methods
resulted in more than 8100 protein groups, with a lower median
coefficient of variation between replicates for Anl-enrichment sam-
ples, indicating better quantitative precision (Fig. 2c, d). While around
70% of the identified protein groups overlapped between both FACS
and Anl-enrichment-based analysis (Fig. 2c), exclusive identifications
with each method revealed distinct strengths. Flow cytometry-sorted
samples showed, for example, enrichment of transmembrane pro-
teins, likely facilitated by strong ionic detergent-based lysis, which
enhances transmembrane protein extraction and digestion[41,42] but can
interfere with CuAAC reactions[34]. Conversely, cell-selective labeling
captured proteins released by cells, such as ECM components and
cytokines, specifically well (Fig. 2e). We primarily attribute this to the
enrichment of proteins from the interstitial space in tumors, which are
accessible for MetRS\*-based cell-selective proteomics but lost in
tissue-dissociation and sorting-based protocols.

## Co-culture promotes inflammatory responses of PDAC cells and polarization of primary macrophages

After closing the gap to state-of-the-art proteomics performance and
extending Anl labeling applications to in-depth cell-selective secre-
tomics, we applied our toolkit to study pancreatic cancer biology. Both

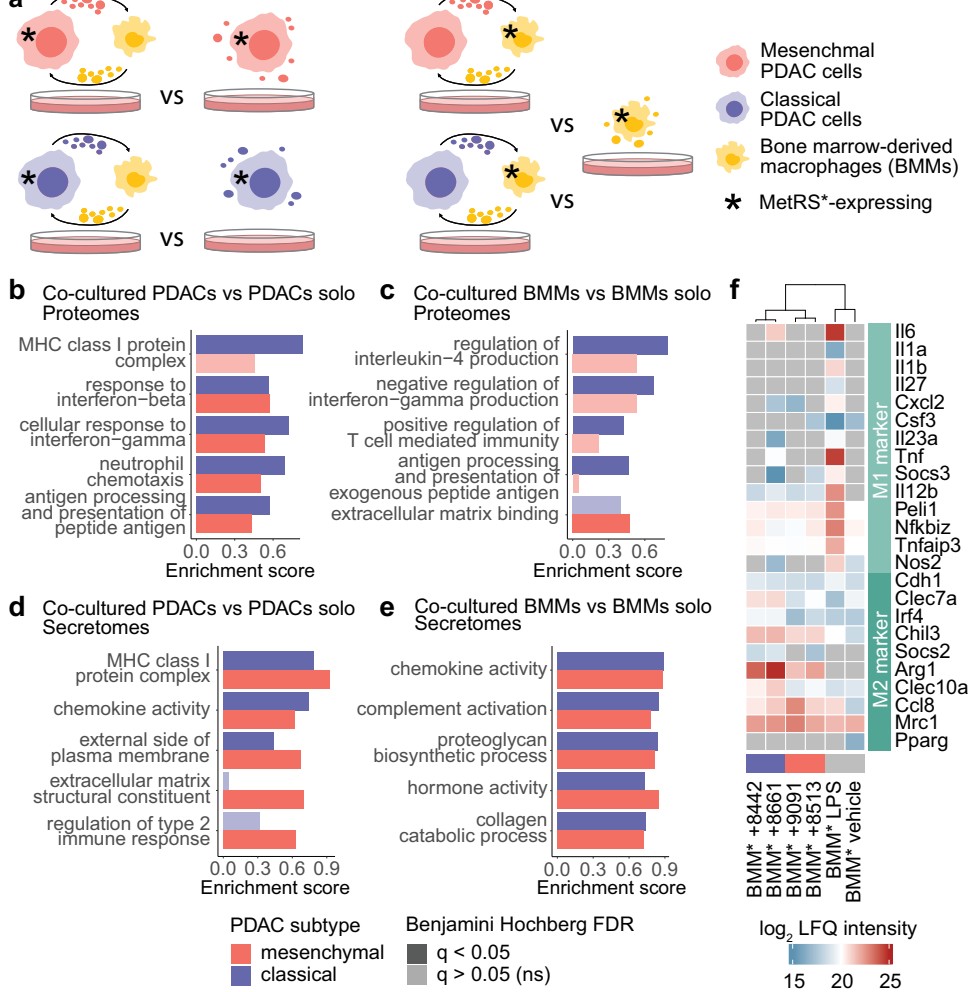

**Fig. 3 | Co-culture of cancer cells with primary macrophages induces bidirectional adaptions. a** Scheme of PDAC and primary macrophage cells in co-culture. Wild-type (WT) or MetRS*-expressing mesenchymal (8513 and 9091) or classical (8442 and 8661) PDAC cells and BMMs were cultured in isolation or co-culture for 36 h with Anl labeling during the last 8 h ($n = 3$, workflow replicates). Asterisks indicate MetRS* expression. **b–e** Strongly enriched gene ontology (GO) terms and UniProtKB keywords in upregulated proteins after PDAC−BMM co-culture compared to each cell type in isolation (two-sided 1D annotation enrichment[128] (full list in Supplementary Data 4)). **f** Heatmap of protein intensities associated with macrophage polarization states in BMM proteomes and secretomes.

the coverage of lower abundant proteins and cell type-resolved information on released signaling proteins are invaluable for understanding intercellular communication. In cancer, complex interactions between transformed cells and tumor stromal cells shape each other's phenotypes and the overall tumor biology. Macrophages, for example, are a major component of solid tumors and are among the earliest tumor-infiltrating immune cells in PDAC[43,44]. To evaluate the potential of MetRS*-based cell-type specific proteomics for the molecular dissection of such intercellular crosstalk, we explored the bidirectional interaction between PDAC cells and macrophages in a controlled in vitro setting. All primary PDAC cell cultures were derived from a genetically engineered $Kras^{G12D}$-driven autochthonous mouse PDAC model[45]. They are representative of the classical subtype, displaying an epithelial morphology ("8661" and "8442"), or of the basal-like mesenchymal subtype ("8513" and "9091"), characterized by increased oncogenic *Kras* gene dosage (*Kras-mut* iGD) and a particularly unfavorable prognosis. By generating LysM-Cre-MetRS* mice, which specifically express MetRS* in the myeloid compartment, we were able to obtain primary MetRS*-expressing bone marrow-derived macrophages (BMMs). We then cultured the four PDAC lines, and the BMMs alone or in co-culture (Fig. 3a) and cell type-selectively analyzed proteins from cells and cell supernatants. Principal component

analyses (PCAs) showed reciprocal adaptions of cancer cells and BMMs to co-culture with changes in both global proteome expression and protein secretion, although less clear for PDAC secretomes (Supplementary Fig. 6). PCAs further indicated distinct differences between PDAC subtypes and PDAC line-specific BMM responses.

We first investigated broad trends and processes in the proteome and secretome dynamics between each cell type in isolation and co-culture. A gene ontology (GO)[46] enrichment analysis showed increased expression of antigen-presentation and major histocompatibility complex (MHC) class I-associated proteins in classical, and to a lesser degree in mesenchymal PDAC cells (Fig. 3b), which was previously observed in breast cancer cells co-cultured with macrophages in transwell systems[47]. Interaction with BMMs also induced strong upregulation of chemokine production and interferon response signatures in both classical and mesenchymal PDAC cells, while, in particular, mesenchymal cells strongly increased structural matrix protein deposition (Fig. 3b, d). Secretomics analysis at the individual protein level revealed secretion of complex immunomodulatory signals with pronounced differences between PDAC subtypes and significant changes upon interaction with macrophages (68 signaling proteins with cytokine function and significant abundance differences (ANOVA, FDR = 5%, S0 = 0.1) between subtypes and culture conditions, see

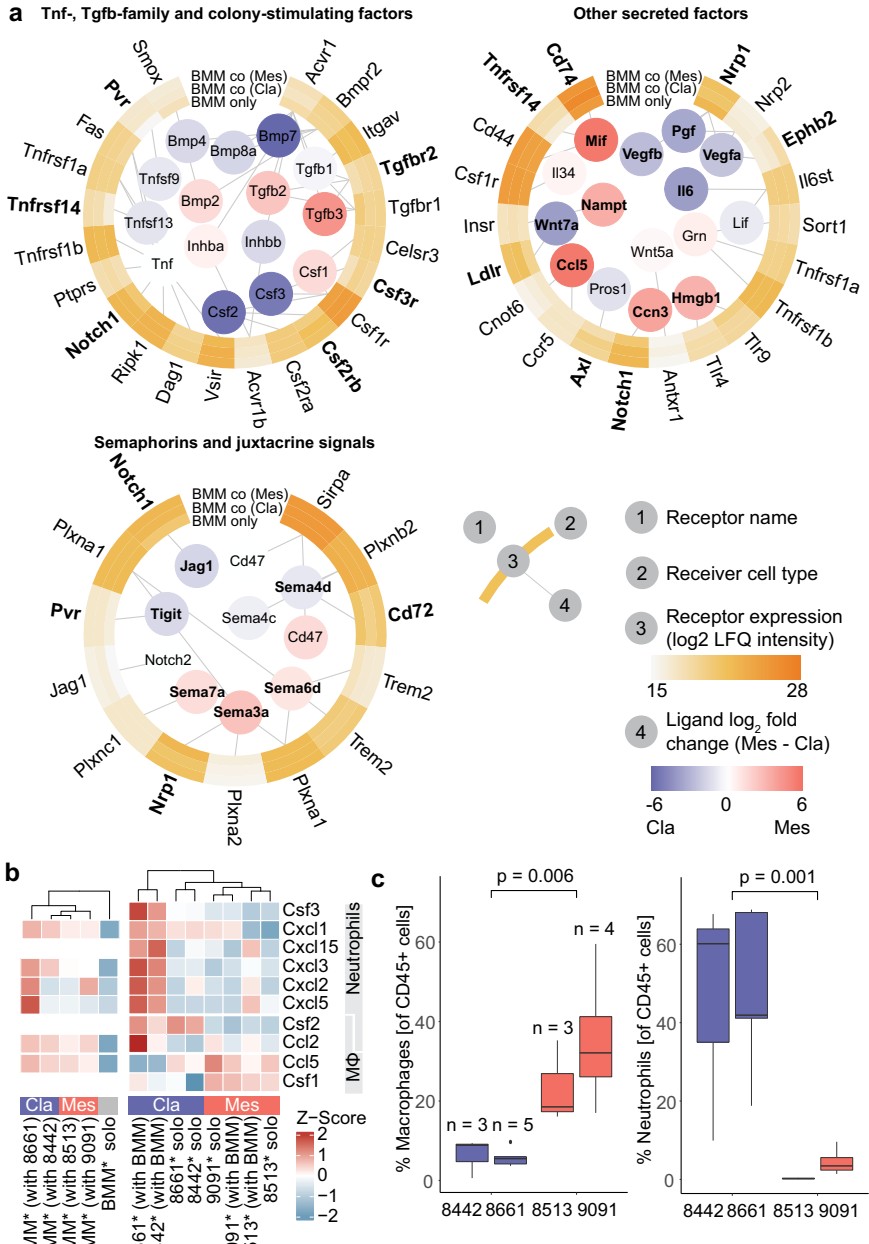

**Fig. 4 | Intercellular signals and signal receptors involved in BMM adaptions to PDAC co-culture. a** Macrophage polarization-associated intercellular signaling protein expressed by mesenchymal or classical PDAC cells in co-culture with corresponding receptors detected in BMM proteomes. Ligands with significantly different secretion among PDAC subtypes (two-sided Student's *t*-test, permutation-based FDR = 0.05, S0 = 0.1) and BMM receptors with significantly different levels of abundance between culture conditions (ANOVA, permutation-based FDR = 0.05,

S0 = 1) are indicated in bold. **b** Heatmaps of primarily macrophage and neutrophil chemoattractants secreted by PDAC cells and BMMs. **c** Macrophage and neutrophil count as a percentage of CD45-positive cells in tumors from orthotopically transplanted classical and mesenchymal cancer cells analyzed by flow cytometry. Numbers of biological replicates and two-sided Welch's *t*-test *p*-values between cell ratios in classical or mesenchymal tumors are indicated. Source data are provided as a Source Data file.

Supplementary Fig. 7). For example, co-culture induced increased Il6 release by both subtypes but with much higher levels in classical PDAC cells, whereas specifically mesenchymal PDAC cells strongly increased secretion of CCL8 and 9. Moreover, significant enrichment of surface-exposed plasma membrane proteins in secretomes, including MHCI proteins, suggested increased shedding activity in cancer cells (Fig. 3d).

Upon interaction with classical PDAC cells, BMMs expressed higher levels of proteins associated with exogenous antigen presentation, T cell regulation, and regulation of key cytokines involved in the coordination of pro- and antitumoral response reactions[48,49]

(Fig. 3c). Although trends could also be observed upon co-culture with mesenchymal PDAC cells, effects were less pronounced and did not reach statistical significance. However, BMM secretomes showed strong enrichment of immunomodulatory proteins, hormones, and growth factors, and extracellular matrix (ECM)-modifying proteins after co-culture with both PDAC subtypes (Fig. 3e). In addition to many cytokines, interaction with cancer cells broadly induced ECM regulators such as matrix metalloproteinases (MMPs), a disintegrin and metalloproteinases (ADAMs), and ADAMs with thrombospondin motifs (ADAMTSs), with crucial functions in cancer[50,51] (Supplementary Fig. 7).

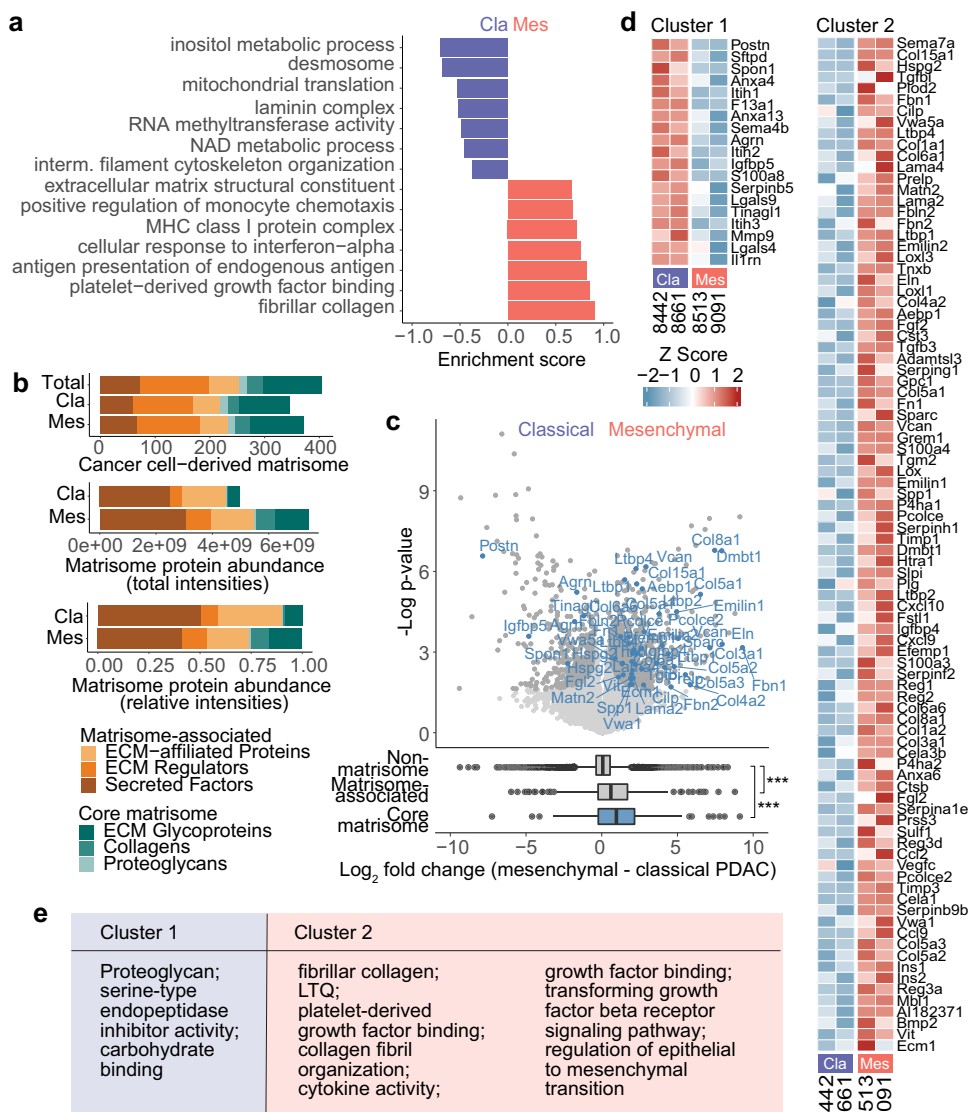

**Fig. 5 | PDAC subtype-specific expression of cancer cell-derived matrisome proteins in primary tumors. a** Significantly enriched gene ontology terms (1D annotation enrichment, Benjamini–Hochberg FDR = 0.05) in classical and mesenchymal cancer cell proteomes after orthotopic transplantation into fully immunocompetent syngeneic mice (8442, 8513, 8661: n = 3, 9091: n = 2, biological replicates) (full list in Supplementary Data 4). **b** Left: Cancer cell-derived matrisome proteins annotated accoding to Naba et al.[75] Right: Summed total and relative LFQ intensities per matrisome category. **c** Volcano plot of cancer cell-derived proteins in mesenchymal and classical PDAC tumors. Significantly regulated core matrisome proteins are highlighted in blue (two-sided Student's t-test, permutation-based FDR = 0.05, S0 = 0.1). Boxplots show quantitative distributions of core matrisome, matrisome-associated and all identified non-matrisome proteins between the PDAC subtypes. P-values were determined by two-sided Welch's t-test: ***$p \le 0.001$ (1: $p = 1.7 \times 10^{-5}$, 2: $p = 5.9 \times 10^{-6}$). **d** Heatmap of matrisome proteins with significant expression differences between classical and mesenchymal PDAC cells in vivo (two-sided Student's t-test, permutation-based FDR = 0.05, S0 = 0.1). **e** Gene ontology terms and UniProtKB keywords found overrepresented by Fisher's exact test within each cluster are indicated (Benjamini–Hochberg FDR = 0.05, full list in Supplementary Data 4). Source data are provided as a Source Data file.

## Co-cultured macrophages acquire TAM-like features driven by a complex mix of cancer cell-secreted and -displayed signaling proteins

To further evaluate the cancer-cell-induced macrophage states in co-culture, we annotated proteins from BMM-selective secretome and cellular proteome datasets using a panel of markers commonly associated with macrophage polarization[52]. M1 and M2 states exemplify broadly clustered extremes on a spectrum of macrophage states—with M1 being associated with interferon and Toll-like receptor signals and efficient production of effector molecules and inflammatory cytokines, and M2 macrophages with the resolution of inflammation or $T_H2$ response-driven physiological reponses[52,53]. As an experimental reference, we stimulated MetRS*-expressing BMMs with lipopolysaccharide (LPS), a Toll-like receptor 4 (TLR4) agonist, and acquired their proteome and secretome profiles. As expected, LPS-stimulated BMMs showed exclusive expression and strong upregulation of M1-associated marker proteins compared to unstimulated cells (Fig. 3f). PDAC co-cultured macrophage M1-associated marker expression was detected only sporadically and mostly at basal levels, except for increased secretion of the pro-inflammatory cytokines Il6 and Tnf upon interaction with 8661 classical PDAC cells. Instead, co-culture primarily induced upregulation of M2-associated markers such as Arg1 and Chil3, again often with stronger responses to classical PDAC cells. Together, cancer cell co-culture therefore induced fast and profound adaptions in BMMs reminiscent of tumor-associated macrophage (TAM) features, which often show M2-like differentiation, contribute to immune cell recruitment and regulation, and remodeling of the tumor ECM[54–56].

Cell-selective proteomes and secretomes also allowed us to investigate potentially active intercellular signaling circuits that shaped macrophage polarization. We mapped PDAC-released proteins with corresponding macrophage receptors using ligand-receptor interactions curated in CellPhoneDB[39]. While PDAC cells did not release hallmark M2 polarizing factors IL-4 and IL-13, we found a complex mix of other proteins that have been associated with macrophage polarization (Fig. 4a): Some proteins were secreted without significant differences between subtypes, such as Tgfb1, a known M2 promoter[57], or Tnf (at much lower abundance, see also Supplementary Data 1), an important M2-suppressing factor in cancer[58]. Other Tgfb- and Tnf-family members, macrophage survival essential colony-stimulating factors (Csfs), and many other signaling proteins, however, showed strong and consistent differential expression between PDAC subtypes. Classical PDAC cells secreted, for example, more Il6, a pleiotropic cytokine that has been described to enhance both M1- or M2-like states[59], and Tnfsf15 (Supplementary Fig. 7), which has recently been shown to promote macrophage differentiation toward an M1 phenotype and increased cancer cell phagocytosis[60]. Mesenchymal PDAC cells secreted higher levels of Tgfb2 and Tgfb3 as well as Mif, Ccl5, and the alarmin Hmgb1, which have been found to skew macrophage polarization toward M1- and M2-like states in a context-dependent manner[61–64]. Furthermore, both PDAC subtypes expressed similar levels of Cd47, a contact-dependent anti-phagocytotic signal often upregulated by cancer cells to escape elimination by phagocytes[65,66]. Also, there is increasing evidence for the contribution of semaphorins to macrophage recruitment and differentiation, a family of exclusively secreted (class 3) or membrane-bound proteins (that can act as contact-dependent signals) with important roles in cancer[67]. For example, increased Sema3a has been associated with poor outcomes in PDAC[68] and attraction of tumor-associated macrophages[69], while Sema7a was shown to recruit and polarize macrophages toward the M2 state in the context of sepsis[70].

The majority of detected PDAC signal corresponding receptors on BMMs showed stable expression, but some were regulated upon co-culture with cancer cells (Fig. 4a). Notably, BMMs upregulated Pvr (Poliovirus receptor) expression after interaction with both PDAC subtypes. Pvr activation on macrophages has been linked to an anti-inflammatory phenotype[71], and targeting the Pvr-Tigit axis is being explored as a potential cancer immunotherapy strategy[72].

## PDAC cancer cell subtype-specific chemokine secretion patterns correlate with immune cell recruitment in vivo

Overall, both macrophages and PDAC cells responded to co-culture with increased production and release of immunomodulatory signaling proteins. Looking specifically at expression differences of immune cell recruiting factors, we noticed clear trends between PDAC subtypes: Mesenchymal PDAC cells secreted high levels of key monocyte recruitment and macrophage survival signals such as Ccl5 and Csf1 (Fig. 4b). In isolation, all four PDAC cell lines secreted many primarily neutrophil attracting proteins at similar levels. Interestingly, interaction with BMMs, however, strongly induced neutrophil recruiting chemokines like Cxcl2, Cxcl3, Cxcl5, and Cxcl15 in classical PDAC cells, whereas release remained unchanged (9091) or increased much less (8513) in mesenchymal cancer cells. BMM chemokine secretion patterns followed similar trends. Intrigued, we investigated the TME composition of tumors formed by the four PDAC subtype lines after orthotopic transplantation into mice. Immunophenotyping by flow cytometry revealed subtype-specific differences in immune cell populations. Among the analyzed cell types, differences between macrophage and neutrophil recruitment were the most significant and reflected the recruitment factor expression patterns from our secretomics experiments (Fig. 4c and Supplementary Fig. 8).

## PDAC tumors show systematic differences in matrisome protein production between mesenchymal and classical cancer subtypes

To further investigate subtype differences between classical and mesenchymal PDAC, we transplanted all four MetRS*-expressing cancer lines orthotopically into syngeneic mice and compared cancer cell protein expression in the complex TME in vivo. In total, we identified 9415 specifically enriched cancer cell-derived proteins, which makes this one of the deepest cell type-specific PDAC in vivo proteomics datasets to date. Gene ontology enrichment analysis indicated pronounced differences in hallmark processes of epithelial–mesenchymal transition (EMT), such as cytoskeleton organization, ECM modulation, and cell–cell junctions (Fig. 5a). Moreover, mesenchymal PDAC cells showed an enriched interferon response signature and elevated antigen presentation-related protein expression, reminiscent of the adaptions that we observed in co-culture with macrophages in vitro and coinciding with the higher macrophage infiltration in these tumors (Fig. 3b and Fig. 4c).

Notably, many of the most prominent differences between mesenchymal and classical PDAC cell protein expression in vivo were ECM-related. Among diverse functions in cancer progression, dysregulated ECM in tumors strongly contributes to drug resistance, immune suppression, and metastasis[51]. Recent research has shown that, in particular, pancreatic cancer cell- rather than stromal cell-derived matrix proteins correlate with poor patient survival, although contributing only a minor fraction of the total ECM mass[73,74]. This introduced cell type-resolved profiling of ECM in tumors as a promising resource for therapeutic target and biomarker discovery. In contrast to the previous studies that characterized the cancer cell-derived matrix using xenotransplants[73,74], Anl-labeling allows cell type-resolved analysis in syngeneic immunocompetent mice. Therefore, our PDAC model integrates interactions with infiltrating immune cells, which directly modulate the tumor ECM and change the ECM-associated protein expression of other cell types such as cancer cells[50,51] (see also Supplementary Fig. 7). Motivated by this and the previously demonstrated advantages of Anl-enrichment for extracellular protein characterization (Fig. 2e), we further investigated ECM-related proteins in our data.

We annotated proteins that constitute the ECM using an in silico defined matrisome atlas by Naba et al.[75], which specifies "core matrisome" proteins such as collagens and proteoglycans, or proteins that are "matrisome-associated" such as ECM remodeling enzymes or secreted growth factors and cytokines that are known to bind to the ECM. Cancer cells expressed a diverse representation of each category, covering 405 matrisome proteins with only minor differences in overall identification numbers and very similar class distribution between classical and mesenchymal subtypes (Fig. 5b). Mesenchymal PDAC cancer cells have been shown to suppress cancer-associated fibroblasts (CAFs), the most prominent producers of ECM proteins in PDAC tumor stroma, leading to tumors with lower overall stromal and collagen content than classical PDAC[76]. However, quantitative analysis of cancer cell-derived proteins showed a higher abundance of mesenchymal-derived matrisome proteins and over-proportional expression of core matrisome and ECM regulators (Fig. 5b), indicating an increased relative contribution to the tumor ECM. Rather than being driven by a few highly abundant outliers, increased abundance of core matrisome expression in mesenchymal cells was a broad and statistically significant motif (Fig. 5c).

At the individual protein level, more than a hundred matrisome protein groups had significant expression differences between the two PDAC subtypes (Fig. 5d). This included proteins recently identified as promising therapeutic targets, such as the predominantly cancer cell- rather than stromal cell-expressed PDAC metastasis promoters Agrn, Serpinb5, and Cstb[74]. All three proteins were detected in our experiment, and classical PDAC cancer cells produced significantly more

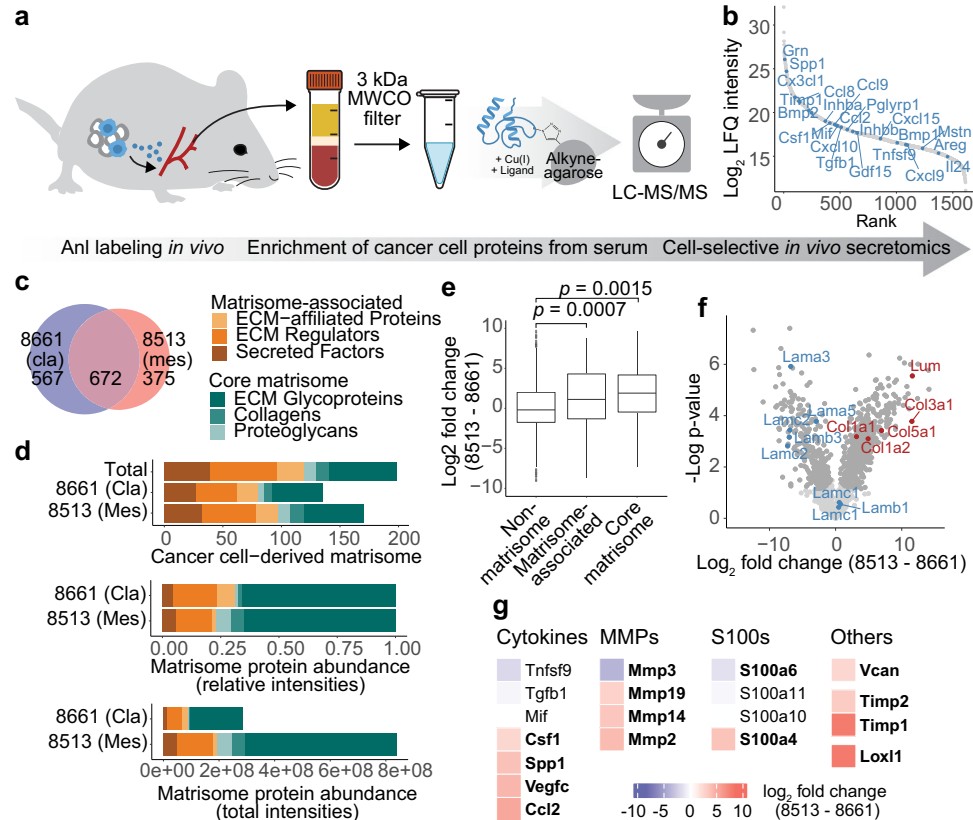

**Fig. 6 | PDAC subtype-specific detection of cancer cell-derived proteins in circulation. a** Scheme of cancer cell-derived protein capture from serum: After orthotopic transplantation of MetRS*-expressing or WT (Ctrl) PDAC cells and ANL labeling, tumor-bearing mouse sera were collected, and tumor-derived proteins were enriched and analyzed. **b** Specifically enriched cancer cell-derived serum protein groups ranked by intensity. Proteins with cytokine function are indicated. **c** Exclusively identified and overlap of specifically enriched 8513 and 8661 cancer cell-derived proteins (8661: $n = 4$, 8513: $n = 3$, biological replicates). **d** Cancer cell-derived matrisome proteins counts, summed total and relative LFQ intensities per matrisome category. **e** Fold change distribution of non-matrisome, and matrisome class proteins between PDAC subtypes. *P*-values were determined by a two-sided Welch's *t*-test. **f** Volcano plot of 8661 and 8513 cancer cell-derived proteins in serum. GOCC annotated Laminin complex proteins (blue), fibrillar collagens (red), and proteins with significant fold changes (dark gray) are highlighted (two-sided Student's *t*-test, permutation-based FDR = 0.05, S0 = 0.1). **g** Fold changes of pre-metastatic niche formation-associated proteins. Proteins with significant fold changes (two-sided Student's *t*-test, permutation-based FDR = 0.05, S0 = 0.1) are indicated in bold. Source data are provided as a Source Data file.

Agrn and SerpinB5, suggesting potential subtype-specific responses to inhibition. Mesenchymal cancer cells, however, consistently produced higher amounts of EMT-promoting matrisome components, for example, fibrillar collagen types I and V, fibronectin, Fgf2, Tgfb family proteins such as Tgfb3 and Bmp2 as well as proteins involved in Tgfb signaling modulation, indicating a feed-forward loop with sustained local EMT signals (Fig. 5d, e). Moreover, we detected a much higher expression of lysyl oxidases Lox, Loxl1, and Loxl3 in mesenchymal cancer cells. Lysyl oxidase-family members mediate crosslinking of collagens and elastin and regulate cellular processes like adhesion, motility, and invasion[77]. They correlate with unfavorable patient prognosis in many cancers, including PDAC, and have been shown to promote chemoresistance, EMT, and metastasis[78].

### In vivo, secretomics reflects tumor subtype and pathogenesis based on more than 1000 cancer cell-derived proteins in circulation

In addition to local effects, tumor cell-derived proteins can act in distant tissues after entering circulation through the lymph or leaky blood vessels. Contrary to inference from cell expression data, profiling of such proteins in the bloodstream would give spatially-specific insights into many crucial aspects of tumor progression that involve long-distance signals and effectors and would also be invaluable for biomarker discovery. However, the lack of cell type-selectivity and the high dynamic range, with extremely abundant functional blood

proteins and comparatively low abundance of tissue leakage proteins[79], make this challenging with conventional methods.

To evaluate whether Anl-labeling could be used to enrich cancer cell-derived proteins directly from body fluids, we collected mouse serum after orthotopic PDAC transplantation and Anl labeling of MetRS*-expressing cancer cells (Fig. 6a). Particularly, serum samples from mice bearing the 8661 (classical) and 8513 (mesenchymal) PDAC subtype tumors showed a good signal-to-noise ratio, with all replicates distinctly clustering from negative controls and each other in a principal component analysis (Supplementary Fig. 9). In these samples, 1614 proteins passed filtering criteria for specific enrichment, including 64 CellPhoneDB-annotated intercellular signaling protein ligands such as 23 cytokines (Fig. 6b), with around 42% identification overlap between the subtypes (Fig. 6c).

After discovering significant differences in matrisome protein expression between both PDAC subtypes in primary tumors, we were interested if these would be reflected in circulation. In total, we detected 199 core matrix or matrix-associated cancer cell-derived proteins in serum. While identified proteins had a very similar qualitative matrisome class distribution as primary tumors, quantitative distribution in serum was distinct, with the top two highest summed intensity classes shifting from secreted factors and ECM-affiliated proteins to ECM glycoproteins and ECM regulators (Figs. 6d and 5b). However, differences between subtypes followed the trends observed in primary tumors: Mesenchymal-derived matrisome proteins were

more abundant and had an over-proportional share of core matrix proteins, specifically collagens and proteoglycans (Fig. 6d, e). At the individual protein level, we again found a higher abundance of laminin complex proteins in classical and fibrillar collagens in mesenchymal PDAC samples (Fig. 6f). Also, key subtype differences in immunomodulatory and matrix-modifying protein release identified in our previous experiments could be captured in serum, such as high Loxl1 and Csf1 secretion by mesenchymal cancer cells, providing direct evidence for potential long-range effects (Fig. 6g). Both proteins have indeed been described to pre-condition future places of metastases and, strikingly, we identified many other previously described pre-metastatic niche conditioning factors[80–83] (Fig. 6g). A supportive pre-metastatic niche is vital for metastatic colonialization, which is considered a rate-limiting step of the invasion-metastatic cascade[84,85]. The premetastatic-niche-promoting signature of cancer cell-derived proteins in circulation likely contributes to the previous observation that increased Kras[G12D] gene dosage (Kras mut-iGD) promotes EMT and metastasis in the mesenchymal PDAC subtype[45].

## Discussion

Increasing insight into intercellular communication in the tumor microenvironment has helped to identify cancer vulnerabilities, for example, crucial immunosuppressive intercellular signaling circuits[40,86]. The combination of MS-based proteomics and cell-selective labeling is emerging as a powerful strategy to further accelerate the knowledge gained about heterocellular processes driving the disease. Direct analyses of labeled peptides offer straightforward solutions for multiplexed cell type-resolved proteomics and the evaluation of enrichment specificity[87]. However, with thorough background interference controls, analyses of all peptides from labeled proteins yield increased sensitivity and protein quantification accuracy.

Nonetheless, previous studies were limited by low proteome coverage and/ or used extensive offline fractionation and less stringent filtering criteria to increase identifications[19,23,28,88,89]. The latter comes at the cost of sample throughput and specificity of enriched proteins, which are both key for the discovery of specific pathophysiological mechanisms. Here, we strongly increased the achievable proteomics depth with Anl labeling-based cell-selective proteomics by improving the biochemical enrichment of azide-modified proteins from complex biomolecule backgrounds. Beyond that, the increased recovery and enrichment specificity enabled additional applications for this concept, where low signal-to-noise was previously prohibitive. Combined with high-end mass spectrometry, data-independent acquisition, and recently developed software[90], our workflow provides comprehensive and MS time-efficient cell-selective proteomes and secretomes in vitro and in vivo.

In this study, we applied our improved workflows for the in-depth exploration of TME features between classical and mesenchymal PDAC subtypes in model systems with different degrees of complexity. In vitro, co-culture experiments offer a very controlled environment for focused and detailed mechanistic investigation of heterocellular interaction. Frequently used indirect co-culture systems such as supernatant transfer experiments or transwell assays facilitate cell-selective analysis by keeping cell types physically separated. However, they cannot cover all communication channels and reciprocal signaling dynamics[91,92]. In contrast, cell-selective labeling enables analysis of cells in direct co-culture, which fully integrates reciprocal communication means, including cell contact formation. Cell-selective labeling using amino acid precursors (CTAP)[93], for example, granted invaluable insight into reciprocal signaling between pancreatic cancer cells and fibroblasts[18]. Specific enrichment of cell-selectively Anl-labeled proteins allowed us to extend this concept to in-depth profiling of heterocellular secretomes. Our findings not only recapitulated the pro-inflammatory secretory programs of macrophages as

determined previously by distinct proteomics methods independent of MetRS*[12,38] but identified, for example, 68 cancer cell-derived proteins with cytokine function in serum-containing culture media. Co-culture of PDAC cells and macrophages underlined the advantage of experiments that allow bidirectional intercellular communication, revealing broad reciprocal adaptions and strong regulation of intercellular signals upon co-culture, with an overall trend toward increased secretion of cytokines and chemokines in both cell types. While macrophages will be exposed to many more stimuli in tumors in vivo, including signals from other stromal cell types, both PDAC subtypes secreted a complex mix of macrophage polarization-associated proteins. Direct interaction with PDAC cells for less than two days was sufficient for macrophages to acquire many TAM-associated features. Moreover, in vitro secretomics allowed in-depth analysis of cancer cell-released chemokines and showed systematic differences between PDAC subtypes that reflected significant differences in TME cell composition, with higher macrophage infiltration in mesenchymal and higher neutrophil infiltration in classical PDAC tumors. This suggests that pancreatic cancer cell-derived signals directly contribute to the recruitment of these cell types. Specifically, large parts of the classical PDAC cell neutrophil recruiting signature became only apparent in co-culture, exemplifying how heterocellular systems expand the intercellular signaling capacity of a single cell type.

For cell type-specific proteomics analysis in vivo, cells are typically extracted from tissue and sorted by FACS or MACS. We have shown that MetRS*-based cell-selective protein labeling and enrichment can have a fundamental cell-type-of-interest protein recovery advantage in pancreatic tumors compared to cell sorting. The high specific yields promise more effective analysis of less abundant or less robust cell types and even provide enough peptides to reach the higher input material demands of extended proteomics techniques such as offline fractionation for the construction of large peptide libraries or post-translational modification-specific enrichment. Importantly, Anl-enrichment also allows freezing of the cell states in tissues directly after harvesting, which provides an additional major benefit for PTM analysis since PTM-states such as protein phosphorylation are often highly dynamic and can be enzymatically modified within minutes in response to environmental changes[94,95]. The combination of Anl-enrichment and PTM analysis, therefore, outlines highly promising avenues for future research.

Here, we focused on another advantage of the technique—the accessibility of extracellular proteins for click chemistry enrichment. Anl labeling facilitates the cell-selective analysis of secreted proteins in tissue or body fluids, which is of great interest and difficult to achieve with conventional techniques. For example, recent pioneering work has demonstrated the high value of cancer cell-selective matrisome analyses in primary tumors and metastases[73,96] but relied on xenotransplants and immunocompromised mice to achieve cell-selectivity. In comparison, MetRS*-based cell-selective proteomics can provide additional value by overcoming the need for species-distinguishing peptides and avoiding potential dynamic range issues caused by the co-analysis of abundant host proteins, which can both reduce the cell-selective matrisome coverage. Moreover, in principle, any cell type can be studied without the need for immunosuppression.

Our MS-based proteomics approach revealed pancreatic cancer cell and subtype-specific matrisome proteins in fully immuno-competent mice and provided proof-of-concept for in-depth analysis of cancer cell-derived proteins in tumor-bearing mouse serum. Previous research has shown a higher cellularity, less activated CAFs, and a less pronounced desmoplastic reaction in mesenchymal PDAC tumors[97]. Our cell-selective tumor analysis revealed that, among the lines we tested, mesenchymal cancer cells themselves produce significantly higher levels of matrisome proteins, particularly core matrix proteins, compared to classical PDAC cells. Furthermore, mesenchymal cancer cells exhibited a distinct matrisome signature

that promotes epithelial-to-mesenchymal transition (EMT). Studies of human PDAC patient cohorts have demonstrated a high ECM content for the mesenchymal subtype, and mesenchymal/ECM-high PDAC correlated with a poor prognosis in comparison to classical/immune-rich PDAC[98]. Matrisome protein release could clearly segregate PDAC subtypes in our experiments, even when analyzing cancer cell-derived proteins in circulation. Remodeled ECM in cancer has been appreciated as critical for tumor progression[51,99]. Early trials directed towards broad depletion of aberrant stroma, however, revealed a dichotomous nature of the ECM and underlined the need for a more precise understanding of stromal components and their role in pathogenesis[100–102]. Using cell type-specific MetRS* mouse models, stromal cell types and their contribution to the tumor ECM can be studied systematically in the future. This combined knowledge will help to evaluate the sources and functions of individual tumor stromal components and identify tumor-promoting candidates for targeted inhibition without simultaneous interference with protective functions. Specifically, we detected elevated expression levels of Lox, Loxl1, and Loxl3 in mesenchymal cancer cells. Loxl2 has been identified as an independent prognostic factor in pancreatic cancer patients associated with poor survival[103,104]. However, anti-Loxl2 mAb treatment in a PDAC transplantation mouse model has caused a significant reduction in matrix content and accelerated tumor growth[105]. Other studies have shown a significant reduction of metastasis, chemosensitization, and prolonged survival after lysyl oxidase inhibition in PDAC[78] or other cancers[106,107]. Future studies should evaluate how cell-type specific lysyl oxidases shape the composition of the tumor microenvironment and contribute to cancer progression.

Cell type-selective profiling of tumor-derived proteins in body fluids opens exciting opportunities for a more precise understanding of long-range intercellular processes such as tumor cell recruitment and the metastatic cascade. In this study, we detected more than 1600 cancer cell-derived proteins in serum, including more than 20 cytokines, strongly improving the coverage achieved in previously published in vivo secretomics approaches, which range from a few dozen to a few hundred cells selectively identified proteins[108–111]. Our data revealed a strong release of pre-metastatic niche formation-associated factors by a mesenchymal compared to a classical PDAC line. Because the abundance of tumor-derived proteins in serum was very low even after enrichment, we expect current developments towards high sensitivity proteomics[112,113] to be highly complementary for even more comprehensive cell-selective in vivo secretomics.

Our study has identified differentiating features among PDAC subtypes with high consistency in our selected models. However, the limited sample size, with only two cell lines per subtype and one line per subtype in the serum secretomics experiment, and the small number of replicates in in vivo mouse experiments do pose limitations to our findings. Despite this, our results demonstrate the unique strengths of cell-selective proteomics analyses in uncovering disease mechanisms and provide a foundation for further research with larger sample sizes to statistically validate and expand upon these findings.

Although our cell type-specific metabolic labeling approach cannot be directly applied to human cancer patients, it offers several possibilities for clinical translation. MetRS* transduced human premalignant cells (e.g., from pancreatic intraepithelial neoplasia (PanIN) or intraductal papillary mucinous neoplasm (IPMNs)), as well as PDAC cells and organoids (e.g., from invasive tumors representing various stages of PDAC progression, differences in metastatic capacity or molecular subtypes), can be transplanted into immunodeficient mice or mice with a humanized immune system[114]. Subsequent MetRS*-based proteomic profiling of tumors and body fluids, such as the blood, enables

not only a deeper understanding of PDAC development, progression, and subtype specification but has also the potential for biomarker identification. So far, biomarkers for PanIN/IPMN and early PDAC detection, subtype classification, prognostic and therapeutic stratification, and the monitoring of targeted interventions are widely lacking[115–117]. Together, MetRS* based proteomic profiling holds the promise of biomarker discovery in tumors and circulation, which can be subsequently tested and validated in prospective studies in cancer patients.

## Methods

### LysMCre-MetRS* mice
*LysM*[Cre/Cre18] and *LSL-R26*[CAG-GFP,Mars*L274G/CAG-GFP,Mars*L274G] (MetRS*)[23] mice have been described previously. Strains were on a C57Bl/6 J background and interbred to obtain homozygous LysMCre-MetRS* mice for bone marrow isolation.

### Cell culture
293 T cells were obtained from ATCC (CRL-3216) and maintained at 37 °C, 5% $CO_2$ in Dulbecco's Modified Eagle Medium (DMEM) supplemented with 10% (v/v) heat-inactivated FCS (FCS HI) (complete DMEM).

Primary mouse PDAC cells were obtained from autochthonous PDAC tumors as described before[119] and maintained in complete DMEM for less than 30 passages.

The preparation of bone marrow-derived macrophages (BMMs) followed the procedure described in Weischenfeldt and Porse (2008)[120]. In brief, bone marrow was harvested from femurs and tibiae of C57BL/6J wild-type (WT) or LysM-Cre-MetRS* mice. Bone marrow cells were passed through 70 μm nylon mesh filters and then plated on sterile, non-tissue culture-treated Petri dishes ($5 \times 10^6$ cells/dish). After culture for 7 days in macrophage differentiation medium (DMEM supplemented with 10% (v/v) FCS HI and 20% (v/v) Csf1-containing L929 cell-conditioned media (replenished on day 3), BMMs were harvested in cold PBS and used for experiments.

Cell lines were authenticated by genotyping and regularly tested for mycoplasma contamination by PCR.

### Transfection, lentivirus production, and transduction
For Met-substitute incorporation comparison experiments, 293 T cells were transfected with an eGFP-MetRS* expression vector based on the pEGFP-C1 (Clontech) plasmid. Transient transfections were done with Lipofectamine 3000 (Invitrogen) according to the manufacturer's instructions.

For stable MetRS* expression, PDAC cells were lentivirally transduced using a modified Precision LentiORF Collection (pLOC) library (GE Healthcare) plasmid (pLOC-CMV > MetRS*:IRES:TurboGFP:P2A:BlastR; enrichment method comparison experiment), generated as described previously[121], or a pLV-EF1A > MetRS*:P2A:EGFP:T2A:Puro plasmid, constructed by VectorBuilder.

For virus production, 293 T cells were transfected with helper plasmids pMD2.G (Addgene), and psPAX (Addgene), and a MetRS* plasmid at a ratio of 1:1.5:2 (3 μg of plasmid DNA in total) in six-well plates using Lipofectamine 3000 and following the manufacturer's instructions for lentiviral production, but using a total of 2 ml complete DMEM for virus collection 48 h post-transfection. After harvesting, polybrene was added to virus-containing supernatants at 10 μg/ml final concentration. 1 ml of virus and polybrene-containing supernatant was added to $2.5 \times 10^5$ PDAC cells seeded in 6-well plates and mixed with 1 ml complete DMEM. After 16 h, media were exchanged with 2 ml complete DMEM. Forty-eight hours post-transduction, successfully transduced cells were selected with antibiotics (Blasticidin for pLOC-MetRS* and Puromycin for pLV-MetRS*) at a final concentration of 10 μg/ml.

## Pulse labeling with azidonorleucine or other Met-substitutes in vitro

Cells were washed twice with PBS and then pre-incubated in methionine (Met)-free DMEM supplemented with 10% FCS HI for 30 min. Afterward, the medium was replaced by Met-free DMEM, 10% FCS HI containing 4 mM azidonorleucine (Iris Biotech) or other Met-substitutes (azidohomoalanine, L-methionine-methyl-$^{13}$C,d$_3$ (Met+4)). Cells were incubated for 8 h, washed twice with PBS, and harvested by scraping. For Anl enrichment-based secretome experiments, serum concentration during labeling was reduced to 5% to avoid protein precipitation in the supernatant concentration steps of the enrichment workflow.

## BMM−PDAC co-culture and LPS stimulation of BMMs

$2 \times 10^7$ primary WT or MetRS* BMMs and $5 \times 10^6$ primary WT or MetRS* PDAC cells were seeded in mono- or co-culture on 15 cm dishes, incubated for 28 h in complete DMEM, and subsequently labeled for 8 h in 4 mM Anl (see above for details). BMMs seeded and cultured in parallel to co-culture experiment samples were treated with 500 ng/ml LPS or vehicle simultaneously with the Anl-labeling. Cells and supernatants were harvested for cell-selective in vitro global proteomics and secretomics analyses (see below for details).

## Orthotopic transplantation and Anl administration in vivo

In vivo transplantation experiments were performed as described in *Nature Cancer volume 3, pages 318–336 (2022)*[40]. In brief, $1 \times 10^4$ MetRS*/WT mouse PDAC cells were orthotopically transplanted into the pancreas of syngeneic immunocompetent C57Bl/6J mice. PDAC cell lines isolated from female endogenous mice were transplanted in female recipients and vice versa for male mice. Two to three weeks after transplantation, mice were treated with Anl (200 µl 300 mM Anl, twice a day for 5 days, intraperitoneal injections). Animals were sacrificed when individual mice reached the human endpoint or after Anl treatment.

All mice experiments were performed in compliance with the European and the ARRIVE guidelines for the care and use of laboratory animals and were approved by the Institutional Animal Care and Use Committees (IACUC) of the local authorities of Technische Universität München and the Regierung von Oberbayern. A tumor diameter of 1.5 cm and a specific burden score, defined by a cumulative burden score, allowed by the IACUC and Regierung von Oberbayern were not surpassed in this study. All mice were kept in dedicated facilities, with a light–dark cycle of 12:12 h, housing temperature between 20 and 24 °C, and relative air humidity of 55%.

## Serum collection

Blood from orthotopically transplanted mice was collected from the submandibular vein in serum collection tubes and further processed for downstream analysis (see below).

## Flow cytometry analysis and FACS

### Acquisition of eGFP-positive PDAC cell cultures by flow cytometry.
Cultured MetRS* and WT 8661 PDAC cell lines were detached using trypsin, then washed three times with ice-cold PBS, filtered through a 30 µm mesh, and resuspended in an adjusted volume of ice-cold PBS. Cell acquisition was performed using the BD FACS Aria Fusion. Flow cytometry data were analyzed using FlowJo software (v10.6.2).

### Acquisition and sorting of eGFP-positive cells from in vivo tumors by flow cytometry.
Dissociation of fresh tumor samples was performed as described previously[40]. Next, the debris removal solution (Miltenyi #130-109-398) was used to discard cell debris from the cell suspension, and the removal of dead cells was performed using the dead cell removal kit (Miltenyi #130-090-101). The enriched fraction of live cells were collected in ice-cold 2% FCS/PBS buffer and filtered

through a 30 µm mesh before acquisition. Cell sorting was performed using the BD FACS Aria Fusion. EGFP-positive cells were sorted in low-bind tubes in PBS, washed two times with PBS, and the resulting cell pellet was shortly dried and snap-frozen. Flow cytometry data were analyzed using FlowJo software (v10.6.2).

**Immunophenotyping by flow cytometry.** Dissociation of fresh tumor samples and antibody staining was performed as described previously[40]. Cells were blocked with anti-mouse CD16/CD32 FC block (Biolegend, 1:100) for 10 min on ice and stained with Zombie Aqua Fixable Viability Kit (Biolegend, 1:500) to discriminate live and dead cells. The following antibody cocktails were used: CD4 BUV805 (BD, 1:100), CD3εBUV395 (BD, 1:20), CD8a BV785 (Biolegend, 1:100), CD25 BV650 (Biolegend, 1:50), TCRγ/δ BV421 (Biolegend, 1:100), CD62L PE (Biolegend, 1:500), CD44 APC-Fire (Biolegend, 1:30), CD45 PerCP Cy5.5 (Biolegend, 1:100), CD19 FITC (Biolegend, 1:100), EpCAM APC/AF647 (Biolegend, 1:200) for acquisition of adaptive immune cells; CD11c BUV737 (BD, 1:30), NK1.1 BUV395 (BD, 1:25), Ly6C BV785 (Biolegend, 1:200), CD11b BV650 (Biolegend, 1:100), F4/80 BV421/PB (Biolegend, 1:30), CD45 PerCP Cy5.5 (Biolegend, 1:100), Ly6G PE (Biolegend, 1:200), CD68 APC-CY7 (Biolegend, 1:20), EpCAM APC/AF647 (Biolegend, 1:200) for acquisition of innate immune cells. $1 \times 10^6$ events were acquired per antibody panel on the BD LSRFortessa. Flow cytometry data were analyzed using FlowJo software (v10.6.2).

## Enrichment of Anl-containing proteins

**DST-based enrichment.** DST-based enrichment was done as described in *Methods in Molecular Biology volume 1266 pages 199–215 (2015)*[122] with slight modifications. In brief, samples were lysed in 1% (w/v) SDS, 2% Triton X-100, PBS pH 7.8 supplemented with EDTA-free protease inhibitors (PI) (Roche), diluted with PBS PI 1:1 for DNA digestion by benzonase (added 1:1000 (v/v)), heated for 10 min at 95 °C, diluted further with PBS PI to a final concentration of 0.1% SDS and 0.2% Triton X-100, and cleared by centrifugation for 5 min at $3000 \times g$, 4 °C. Lysates were reduced and alkylated with immobilized TCEP and iodoacetamide (IAA), and subsequently desalted with PD-10 columns (GE Healthcare) to remove excess of reduction and alkylation agents. Click reactions were started by sequential addition of 200 µM tris((1-benzyl-4-triazolyl)methyl)amine, 25 µM disulfide biotin alkyne-tag (DST) (Click Chemistry Tools), and 100 µg/ml Cu(I)Br suspension and samples were incubated in an end-over-end mixer overnight at 4 °C. Following a second desalting step with PD-10 columns, elution in 10.5 ml 0.05% SDS, PBS pH 7.5, and the addition of 1% (v/v) NP40, tagged proteins were bound to 300 µl washed NeutrAvidin agarose (Thermo Scientific, 29202) in an end-over-end mixer overnight at 4 °C. Afterwards, the resins were sequentially washed with a total of 36 ml 0.2% SDS, 1% Triton X-100, PBS pH 7.4, then 18 ml PBS pH 7.4, and finally 18 ml 50 mM ammonium bicarbonate. Tagged proteins were eluted in a two-step procedure with a 5% (v/v) 2-mercaptoethanol/ammonium bicarbonate solution and subsequently lyophilized. After drying, proteins were resolubilized in 8 M Urea, 50 mM Tris-HCl pH 8 with 1 µg trypsin and lysC, predigested for 4 h at room temperature, and then diluted with 50 mM Tris-HCl pH 8 to a final concentration of 2 M urea for overnight digestion. Digests were desalted with C18 SepPak cartridges and in-house-made styroldivinylbenzol reversed phase sulfonate (SDB-RPS) (3 M Empore, 2241) StageTips.

**DBCO-agarose SPAAC enrichment.** DBCO-agarose enrichment was done as described in Mahdavi et al.[19] with slight modifications. In brief, samples were lysed in 1% SDS, 100 mM chloroacetamide, PBS PI pH 7.4, heated at 95 °C for 10 min, sonicated to shear DNA, and centrifuged at $14.000 \times g$ for 30 min. Cleared lysates were incubated for 3 h at room temperature with 100 µl washed dibenzocyclooctyne (DBCO)-agarose in an end-over-end mixer, and unreacted DBCO groups were subsequently quenched for 30 min by addition of 4 mM Anl. Afterward,

bound proteins were reduced with 10 mM DTT for 15 min at 70 °C and alkylated with 40 mM IAA for 30 min at room temperature. The resins were sequentially washed with a total of 40 ml 0.8% SDS in PBS, 40 ml 8 M urea in 100 mM Tris-HCl (pH 8.0), and 40 ml 20% acetonitrile. Washed resins were resuspended in 100 μl 10% acetonitrile, 50 mM ammonium bicarbonate, and bound proteins were on-bead digested at 37 °C overnight with 1 μg of trypsin and lysC. Digests were collected, resins were washed with 500 μl 50 mM ammonium bicarbonate, washes were combined with digests and desalted with C18 SepPak cartridges.

**Alkyne-agarose CuAAC enrichment.** Samples were lysed in gua-lysis buffer (6 M guanidinium chloride, 4% (w/v) CHAPS, 0.5 M NaCl, 200 mM Hepes (pH 8) PI), heated at 95 °C for 5 min, sonicated to shear DNA and centrifuged at 10,000×g for 30 min. Cleared lysates were mixed with 100 μl (enrichment method comparison) or 50 μl (all other experiments) washed alkyne-agarose and diluted with ddH$_2$O and a premixed catalyst solution to a final concentration of 1.5 M guanidinium chloride, 1 mM CuSO$_4$, 6.25 mM BTTAA (Click Chemistry Tools), and 10 mM sodium ascorbate. Samples were incubated at room temperature overnight in an end-over-end mixer. Afterward, resins were washed twice with ddH$_2$O and once with SDS wash buffer (1% (w/v) SDS, 250 mM NaCl, 5 mM EDTA, 100 mM Tris pH 8). After protein reduction with 10 mM DTT for 15 min at 70 °C and alkylation with 40 mM IAA for 30 min at room temperature in SDS wash buffer, resins were sequentially washed with a total of 20 ml SDS wash buffer, 20 ml 20% isopropanol, 20 ml 6 M guanidinium chloride, 100 mM Tris-HCl (pH 8), and 20 ml 20% acetonitrile. Washed resins were resuspended in 100 μl 10% acetonitrile, 2 mM CaCl$_2$, 50 mM Tris-HCl pH 8, and bound proteins were on-bead digested at 37 °C overnight with 1 μg of trypsin and lysC. Digests were collected, resins were washed with 500 μl ddH$_2$O, washes were combined with digests and desalted with C18 SepPak cartridges (enrichment method comparison and in vitro experiments) or Pierce Peptide Desalting Spin Columns (Thermo Scientific) (in vivo experiments).

For Anl-enrichment-based in vitro or in vivo secretomes experiments, 15 ml cell-conditioned media or 400 μl tumor-bearing mouse serum were collected after Anl labeling (if yields from individual animals were lower, serum from multiple mice was pooled to reach the total volume). Conditioned media were centrifuged for 5 min at 1000×g to remove cell debris and supplemented with protease inhibitors. Conditioned media or mouse sera were washed twice with 15 ml 50 mM Tris-HCl pH 8 and concentrated to a volume of 250 μl using Ultracel-3 regenerated cellulose centrifugation filter units with a 3 kDa molecular weight cutoff (Millipore). Samples were mixed 1:1 with gua-lysis buffer, heated for 5 min at 95 °C, 1200 rpm, and then further processed using the alkyne-agarose CuAAC enrichment workflow (see above).

Before lysis and CuAAC Anl-protein enrichment, tissue samples were homogenized to a fine powder with a mortar and pestle in liquid nitrogen.

**Sample preparation for mass spectrometry**
For proteomics analysis without Anl-enrichment, cells were lysed in SDC buffer (1% sodium deoxycholate (SDC), 10 mM tris(2-carbox-y(ethyl)phosphine) (TCEP), 40 mM 2-chloroacetamide (CAA), 100 mM Tris-HCl pH 8.5) heated at 95 °C for 10 min and sonicated to shear DNA. Proteins were digested with trypsin and lysC (1:100 enzyme/protein ratio, w/w) at 37 °C, 1000 rpm overnight. Digests were desalted using in-house-made SDB-RPS StageTips.

Desalted peptides from workflows with or without Anl-enrichment were dried in a vacuum concentrator and resolubilized in 0.1% formic acid. Concentrations were determined using a Nano-Drop spectrophotometer and normalized between samples for equal peptide injection. Negative control (WT) samples for evaluating Anl-

enrichment specificity were adjusted with corresponding volumes to their corresponding MetRS* samples for injections of equal total yield proportions.

For offline high pH reversed-phase fractionation of peptide samples into 16 fractions (Fig. 1e), a spider fractionator was used as described previously[123].

**LC−MS/MS**
Peptide mixtures were analyzed with an EASY-nLC 1000 or 1200 ultrahigh-pressure system (Thermo Fisher Scientific) coupled to a Q Exactive HF (293 T Met-substitution), Q Exactive HF-X (enrichment and acquisition method comparisons) or Orbitrap Exploris 480 (all other experiments) instrument (Thermo Fisher Scientific). Peptides (500 ng injections for Q Exactives or 300 ng for Exploris machines) were separated on 50 cm in-house-made 75 μm inner diameter columns, packed with 1.9-μm ReproSil C18 beads (Dr. Maisch GmbH) at a flow rate of 300 nl min$^{-1}$ and 60 °C maintained by an in-house-made column oven. Offline pre-fractionated samples used for acquisition method comparison (see Fig. 1) were eluted with a binary buffer system (buffer A: 0.1% formic acid; buffer B: 80% acetonitrile, 0.1% formic acid) and a nonlinear gradient starting at 3% buffer B followed by a stepwise increase to 23% in 82 min, 40% in 8 min and a wash-out step for 10 min with an increase to 98% buffer B. Spectra were acquired with a data-dependent Top15 MS/MS method: Full scans (300–1650 m/z, automatic gain control (AGC) target = 3e6, maximum injection time = 25 ms, resolution = 60,000 at 200 m/z) were followed by up to 15 MS/MS scans with higher-energy collisional dissociation (HCD) (AGC target = 1e5, maximum injection time = 25 ms, isolation window = 1.5 m/z, normalized collision energy (nce) = 27%, resolution = 15,000 at 200 m/z). All other samples were analyzed without prefractionation in single shot measurements with a nonlinear gradient starting at 5% buffer B followed by a stepwise increase to 30% in 95 min, 60% in 5 min and a wash-out step for 20 min with an increase to 95% buffer B and subsequent decrease to 5% buffer B. Spectra were acquired with a data-dependent Top15 MS/MS method (as described above, but full scans with maximum injection time = 20 ms and MS/MS scans with max-imum injection time = 28 ms, isolation window = 1.4 m/z) or data-independent acquisition (used for acquisition method comparison (Fig. 1e) and all following experiments) using full scans with a range of 300–1650 m/z (AGC target = 3e6, maximum injection time = 60 ms, resolution = 120,000 at 200 m/z) followed by MS/MS scans with 32 windows (nce = 27%, AGC target = 1e6, maximum injection time = 54 ms, resolution = 30,000 at 200 m/z). Data acquisition was controlled by Xcalibur (version 4.4.16.14, Thermo Fisher Scientific).

**LC−MS/MS data analysis**
DDA MS raw files were processed by MaxQuant[124] (version 2.0.1.0.) using default parameters for orbitrap instruments with 1% FDR at the peptide and protein level, enabling MaxLFQ for label-free quantification. For analysis of Met-substitute incorporation in 293 T cells, Met-Anl, Met-Aha, and Met-Met+4 substitutions were added as variable modifications.

DIA MS raw files were processed by DIA-NN[90] (version 1.8) with FASTA digest for library-free search and deep learning-based spectra, RTs, and IMs prediction enabled. Precursor FDR was set to 1%, and default parameters were used with the following changes: The pre-cursor range was restricted to 300–1650 m/z, and the fragment ion range to 200 – 1650 m/z. The "--relaxed-prot-inf" option was enabled via the command line. Mass accuracies and scan windows were opti-mized for individual experiments as recommended by the developers. MBR was enabled, neural network classifier was set to "double-pass mode," and the quantification strategy to "robust LC (high accuracy)."

Spectra were matched against the human (June 2022, 79,276 entries) or mouse (January 2022, 55,105 entries) UniProt FASTA database.

Raw files were processed in two separate analyses for optimal independence of FACS- and Anl enrichment-based cancer cell-selective proteomics results (Fig. 2). MetRS*/ WT 8661 tumor rawfiles were reprocessed together with samples from other PDAC subtypes for in vivo PDAC cell subtype comparison (Fig. 5). 8661 PDAC (solo) secretome results (Fig. 3) were also used for secretome benchmarking (Fig. 1).

**Evaluation of Anl-enrichment specificity.** Except for enrichment method benchmarking (Fig. 1c, d), data from all Anl-enrichment-based experiments were filtered for specifically enriched proteins before further analysis. To evaluate enrichment specificity, samples were compared to corresponding negative control samples (WT equivalents of MetRS*-expressing cells that were treated equally and processed in parallel) and only proteins that were not identified in controls or had an at least 3fold higher median intensity than in controls were retained. For technical experiments (Figs. 1 and 2), PDAC MetRS* tumor comparison experiments (Fig. 5), and serum secretomics experiments with the PDAC lines 8661 and 8513 (Fig. 6), corresponding WT controls were used in triplicates for each PDAC line. We used aggregated control sample groups for multiple experimental groups in the co-culture experiments (Figs. 3 and 4): Three BMM WT samples were used to control BMM MetRS* samples cultured in isolation. A group of four co-cultured BMM WT + PDAC WT samples (one with each of the four PDAC lines) was used as controls for all BMM MetRS* + PDAC WT co-culture samples. Both solo and co-cultured PDAC MetRS* samples were controlled with the more conservative corresponding co-culture control samples (PDAC WT + BMM WT in triplicates for each of the PDAC lines).

**Statistical analysis.** Bioinformatic analyses were performed with Perseus[125] (version 1.6.10.43) and R (version 4.1.2). Before statistical analysis, quantified proteins were filtered for at least two valid values in at least one group of replicates. The remaining missing values were imputed by random draw from a normal distribution with a width of 0.3 and a downshift of 1.8 relatives to the standard deviation of measured values. Statistical tests and parameters used to evaluate annotation enrichment and significant abundance differences of quantified proteins are specified in the figure legends. For box-and-whisker plots, standard boxplot features (lower quartile, median, upper quartile) were used as defined by ggplot2 version 3.4.0.

**Intercellular communication analysis.** Interactions between PDAC cells and macrophages in co-culture were inferred based on annotated ligand–receptor interactions from CellPhoneDB[39] (v.2.0) extended by proteins with secretomes-derived experimental evidence[126]. BMM receptor expression levels were sourced from global proteomes, PDAC cell ligand expression levels from secretomes for secreted ligands, and global proteomes for membrane-bound ligands after filtering and imputation of missing values (see above).

### Reporting summary
Further information on research design is available in the Nature Portfolio Reporting Summary linked to this article.

## Data availability
The mass spectrometry proteomics data have been deposited to the ProteomeXchange Consortium (http://proteomecentral. proteomexchange.org) via the PRIDE partner repository[127] with the dataset identifier PXD040084, which is publicly available. Source data are provided in this paper.

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

## Acknowledgements
We thank I. Paron, C. Deiml, A. Strasser, J. B. Mueller-Reif, L. Schweizer, S. Steigerwald and P. Skowronek for MS assistance; V. Boeck, S. Ettl, M. Zukowska, A. Kudaliyanage, Y. P. Alvarez Valdivia, and A. Ullrich for technical assistance; T. Viturawong for intercellular communication analysis framework; M. Oroshi for computer and database support; and M. Mann and Marc Schmidt-Supprian for helpful discussions. This work was funded by the Max Planck Society for the Advancement of Science, the German Cancer Consortium (DKTK), the Wilhelm Sander-Stiftung (2020.174.1 and 2017.091.2) and the Deutsche Forschungsgemeinschaft (DFG, German Research Foundation), Cluster of Excellence EXC 2151 ImmunoSensation2, Project-ID 360372040 (SFB 1335), Project-ID 408885537 (TRR 274), Project-ID 432325352 (SFB 1454), Project-ID 450149205 (TRR 333), Project-ID 329628492 (SFB 1321), Project-ID 515991405 (ME 4766/2-1 and SA 1374/8-1), Project-ID 515571394 (SA 1374/7-1), Project-ID 360394750 (SCHO 1732/2-1), Project ID 458890590 (SA 1374/6-1), Project ID 219542602 (SA 1374/4-3), Project-ID 97850925 (SFB854), and Project-ID 362321501 (GRK2413) and the Bundesministerium für Bildung und Forschung (BMBF) Kmu-innovativ 031B0686B.

## Author contributions
J.J.S. performed experiments. J.J.S. developed and implemented bioinformatics methods. J.J.S. and F.M. conceived the data analysis and interpreted the data. SB and CF bred MetRS*-LysMCre mice, performed PDAC cell culture, characterization, and quality control, in vivo mouse experiments, flow cytometry analyses, and FACS. DS analyzed and interpreted data with S.B. and C.F. A.S. contributed to initial mouse experiments, testing of in vivo labeling schemes, and consultation on data analysis methods. D.F. contributed to co-culture experiments and figure design. J.C. prepared communication plot visualizations. S.E. helped with bone marrow-derived macrophage generation and experiments. P.L. and D.D. supplied reagents, initial training, and consultation on Anl-enrichment procedures. H.D. provided advice and supervision of initial experiments. F.M. conceived the study. F.M. and D.S. provided funding and supervised the experiments. J.S.S. and F.M. wrote the paper.

## Funding

## Competing interests
The authors declare no competing interests.
