## [Peer Review File · Nature Communications]

Cell-selective proteomics segregates pancreatic cancer subtypes by extracellular proteins in tumors and circulationREVIEWER COMMENTS

Reviewer #1 (Remarks to the Author): expertise in proteomics method development

Swietlik et al have improved the process of recovering proteins labeled with azidonorleucine (ANL) from a cell specific BONCAT method. This new method is then applied to studies of pancreatic cancer looking at cell co-culture, extracellular matrix, and proteins circulating in mouse serum. With their new process they identify roughly 10,000 proteins which is a significant increase in protein identification numbers over previous ANL studies. The use of cre-loxed MetRS* allows cell specific labeling of proteins via preferential incorporation of ANL over Met into proteins. This process is not as efficient as MetRS incorporation of Met into proteins, and thus raises a few questions. It is unclear if the new procedure is as specific as the authors claim it to be. Two strategies for enrichment are employed in BONCAT where one enriches intact labeled proteins then either elutes proteins off the column or digests proteins off the column. The improvement reported here uses a cleavable disulfide. The second strategy enriches for ANL or AHA labeled peptides from digested proteins and then elutes peptides off the avidin column. An advantage to the first approach is many more peptides per protein are identified, but the disadvantage is identification is not via a labeled peptide consequently you must be sure the negative control is a good control. An advantage to the peptide strategy is identification is based on a labeled peptide, but the disadvantage is Met is not a very frequent amino acid so there may not be very many peptides per protein available. The identification of the labeled peptides makes the certainty of which proteins are newly synthesized or cell specific high. In the Swietlik et al data how many labeled peptides can be identified? The Maccoss lab developed a tool for mining DIA data for specific peptide sequences called PECAN where you could specifically try to find labeled peptides in the data. How many times was the negative control repeated? Was it consistent each time or did the identities of proteins change and by how much, e.g. if you prepare 10 biological replicates of the negative control with a tight protein identification FDR what is the overlap, 50%, 60%, 70%, etc? The reported protein numbers are high and thus the authors need to be rigorous in establishing those numbers are not simply due to non-specific interactions. Proving labeled peptides exist in their data for a statistically significant number of the proteins that were identified (e.g. of the 10K proteins) will go a long way to establish the legitimacy of the approach.

Summary- overall the paper is potentially an important contribution to the growing area of cell specific proteomics. While I'm not an expert in pancreatic cancer it appears, some interesting discoveries were made using this approach. Cell specific proteomics is potentially an important companion to the emerging area of single cell proteomics (SCP). The potential increase in the number of proteins identified in these studies is about 10X beyond what the current capability is for single cell proteomic methods so it will allow the study of single cell types within complex tissue matrices as SCP matures. And cell specific BONCAT allows you to ask different types of questions.

Other questions.

Line 396-397: Is this over 8000 mice?

Reviewer #2 (Remarks to the Author): expertise in pancreatic cancer cell biology

Swietlik et al. describe an improved method to selectively label and profile proteome contents in pancreatic cancer model systems. The system utilizes azidonorleucine to label and subsequently profile proteome using mass spectrometry. The study is well-designed and clearly written and demonstrates differential proteome coverage over the flow cytometry-based dissociation/proteome profiling methods, allowing for detection of some of the less abundant proteins, such as cytokines. The study also demonstrates significant differences in how distinct subclasses of pancreatic cancers, such as basal and classical, regulate myeloid cell polarization and composition of extracellular matrix. Overall, this work is very interesting, although some minor questions remain as to utility of this platform for diagnostic purposes, especially in the setting of human malignancy. Minor points:

1. A demonstration of how well this method can be applied in the human malignancy setting would be of great interest. At the very minimum, there needs to be a discussion of how the authors

envison pipelining such protocols for diagnostic purposes.

2. ECM proteins are typically abundantly expressed in pancreatic cancer. How does the new method compare to other mass spec-based approaches (not based on flow cytometry) in identifying differences in ECM composition? Does the improved coverage in the new method add any information in this regard?

Reviewer #3 (Remarks to the Author): expertise in bioinformatics analysis of proteomics

In this manuscript, Swietlik et al. applied a combination of cell-selective metabolic labeling (MetRS-based Anl labeling) and MS-based proteomics to study pancreatic ductal adenocarcinoma (PDAC) subtypes and their tumoral microenvironment (TME). In a first step, they optimized the labeling and MS analysis workflow for a deep cell proteome and secretome coverage and demonstrated its superiority to cell sorting-based methods in vivo.

In a second part, they applied this optimized protocol to the analysis of PDAC cells from two different subtypes (classical and mesenchymal) co-cultured with macrophages. Using this cell-selective labeling, they were able to selectively study proteomes and secretomes of both PDAC cells and macrophages.

They applied a similar strategy to an in vivo study of cancer cell-derived matrisome proteins through the orthotopic transplantation of Anl labeled classical or mesenchymal PDAC in mice. Finally, they studied PDAC subtypes cell-derived proteins in circulation in the serum of orthotopically transplanted mice.

Although, the techniques used in this study are not new (use of CuAAC for metabolic labeling, combination of MetRS-based Anl labeling with MS proteomics, application of this method in vivo, DIA MS acquisition to improve protein identification), the authors demonstrated their combination to be highly efficient for a cell-selective deep proteome coverage. More innovatively, they applied these techniques to secretome analyses both in culture and in vivo which, to our knowledge, has never been described in the literature so far.

Considering the rising incidence and high lethality of pancreatic ductal adenocarcinoma and the need of a better comprehension of the tumor signaling with its microenvironment, the work presented in this manuscript is of high interest. In addition to provide new insights in the PDAC cell communication with the TME, it also reveals differences between classical and mesenchymal subtypes. Moreover, it demonstrates, through the analysis of serum from PDAC transplanted mice, that this strategy can be used to access cancer cell-derived proteins directly in body fluids which might be essential for the understanding of long-distance signaling in tumor progression and metastasis.

Although, on a statistical point of view, the findings described in this manuscript need to be confirmed by larger studies, this work opens the way to a better comprehension of PDAC subtypes signaling with its TME that may eventually lead to the identification of new biomarkers or therapeutic targets. It also demonstrates the feasibility and usefulness of cell-selective proteomics for short- and long-distance cell signaling studies that may be applied to other types of cancer. Therefore, I believe this manuscript is of great interest to the readers of Nature communications, provided that the authors can address the following points:

- Figure 1c and 1e: although the standard deviations are shown on the figures, it only represents the variability in the total number of protein identifications for each method. It would be interesting to know how many proteins (or which percentage) overlap between the 3 replicates for each method to assess the technical variability. Moreover, coefficient of variation on the protein intensities over the 3 technical replicates should be provided to assess the efficiency of the methods for protein quantification.

- Figure 1g: Considering that a large proportion of signaling proteins have a low molecular weight (e.g., cytokines) and that quantification of proteins on single peptides are usually less accurate. It would be interesting to mention, on the figure or in the text, the proportion of proteins identified with only 1 peptide or with 2 peptides and more.

- Line 199: The authors claimed they doubled the number of protein identifications with their method in comparison to reference #28. However, this comparison is biased by the fact that they

used two different generation of MS instruments (Exploris 480 for this study; Fusion Lumos for ref #28). The authors should remove this statement or discuss the influence of the MS type on the results.

- Line 202-204 and Fig 2e: While it is understandable that flow cytometry method cannot capture cell released proteins, the authors should comment on the reasons why the cell-selective labeling is less efficient in capturing transmembrane proteins.

- Supp Fig 4: For an easier reading of the figure, it would be appreciated to have the mention "classical" or "mesenchymal" next to the number of each cell line.

- Supp Fig 4: This figure would deserve a better description in the text. While it is obvious that the PDAC cell proteomes are modified when co-cultured with BMM, it is not as clear for the secretome. Conversely, both BMM proteomes and secretomes are changing upon co-culture with PDAC cells.

- Figure 3b: It is not clear in the legend if this figure represents GO terms enrichment for proteome data only or for proteome + secretome. By reading the text, it seems to be proteome only but it should be added in the legend. In this case, one can wonder why no GO enrichment analysis has been performed on secretome data.

- Supp Fig 5: The proteins having significant abundance differences should be indicated on the figure (e.g., the 68 cytokine function proteins mentioned line 242).

- Line 268: Define the abbreviations LPS and TLR4

- Fig 3d: On a graphical point of view, the distinction between the 3 yellow bars for the 3 BMM cultures is not clear. In addition, the results do not show a clear relation between the PDAC subtype differential expression of signaling proteins and the expression of their receptors on the macrophage (moreover, it is not even commented on in the text). Therefore, there is a low interest in representing the data in that way and the figure should better focus on the differences between classical and mesenchymal subtypes.

- Line 285-287: Please reformulate this sentence which is unclear.

- Line 296: There is no data (in fig 3d or even in table 1) to support the statement that PDAC subtypes expressed "high levels" of Cd47. The fig 3d only shows it is expressed at similar levels between the two subtypes and table 1 do not show particularly high quantitative values for this protein.

- Line 333: Since, the q-value associated to "MHC class I protein complex" term enrichment in fig. 3b is not significant, it cannot be stated that the MHCI antigen presentation-related protein expression is elevated in mesenchymal PDAC cells in co-cultures.

- Fig 4d: GO terms above the figures could be presented as a table or as a figure of enrichment analysis rather than a simple list.

- Line 413: the word "systematically" should be removed since Lamc1 and Lamb1 do not have higher abundance in Fig 5f.

- Line 494-496: The statement should be moderate on the fact that only one mesenchymal PDAC cell line has been used for the analysis of serum circulating proteins, the second having been left aside based on the PCA in Supp fig 7.

- In general, statistical considerations should be commented on in the discussion: the use of only 3 replicate mice for in vivo experiments, the use of only 2 cell lines per PDAC subtypes, the use of one cell line per subtype in the mouse serum experiment. Although this can be explained by the amount of work it involved, these choices should be explained and the results commented on from a statistical point of view.

RESPONSE TO REVIEWERS' COMMENTS

We thank the reviewers for their valuable comments. We are glad that the reviewers kindly acknowledge the usefulness of our deep cell-selective proteomics and appreciate the insights we generated on cell non-autonomous disease mechanisms by its application to pancreatic cancer.

We also highly appreciate the critical points and constructive feedback.

In our revision, we have performed the suggested experiments and analyses. Specifically, we have performed a series of additional experiments and analyses to address

- 1) the reproducibility of the workflow,
- 2) the comparison with other existing MS methods and
- 3) statistical concerns.

Furthermore, we have edited the text as suggested and incorporated requested changes in several sections, as detailed in our point-by-point response below. Changes are highlighted in the main text.

Reviewer #1:

Swietlik et al have improved the process of recovering proteins labeled with azidonorleucine (ANL) from a cell specific BONCAT method. This new method is then applied to studies of pancreatic cancer looking at cell co-culture, extracellular matrix, and proteins circulating in mouse serum. With their new process they identify roughly 10,000 proteins which is a significant increase in protein identification numbers over previous ANL studies. The use of cre-loxed MetRS* allows cell specific labeling of proteins via preferential incorporation of ANL over Met into proteins.

This process is not as efficient as MetRS incorporation of Met into proteins, and thus raises a few questions. It is unclear if the new procedure is as specific as the authors claim it to be. Two strategies for enrichment are employed in BONCAT where one enriches intact labeled proteins then either elutes proteins off the column or digests proteins off the column. The improvement reported here uses a cleavable disulfide. The second strategy enriches for ANL or AHA labeled peptides from digested proteins and then elutes peptides off the avidin column. An advantage to the first approach is many more peptides per protein are identified, but the disadvantage is identification is not via a labeled peptide consequently you must be sure the negative control is a good control. An advantage to the peptide strategy is identification is based on a labeled peptide, but the disadvantage is Met is not a very frequent amino acid so there may not be very many peptides per protein available. The identification of the labeled peptides makes the certainty of which proteins are newly synthesized or cell specific high.

In the Swietlik et al data how many labeled peptides can be identified? The Maccoss lab developed a tool for mining DIA data for specific peptide sequences called PECAN where you could specifically try to find labeled peptides in the data.

We thank the reviewer for the positive reception of our work and for evaluating our workflow in the larger context of available enrichment methods for Anl labeling-based cell-selective proteomics.

We indeed benchmarked an enrichment method with cleavable disulfide-tags (Fig. 1 a-e). However, our workflow and later biological experiments were based on a direct and covalent capture strategy on alkyne-agarose and subsequent elution by on-resin digestion. In our hands, this straight forward strategy combined with our lysis, click chemistry and clean up procedure has greatly improved the AnI-protein enrichment efficiency compared to alternative workflows, resulting in very high yields and very low levels of unspecific background (see, for example, Fig. 1b). However, like many other techniques that elute enriched proteins by on-resin digestion (such as the popular copper-free DBCO-agarose-based enrichment (Azizian, Nancy G *et al.* (2021), doi:10.1021/acs.jproteome.0c00666, Mahdavi, Alborz *et al.* (2016), doi:10.1021/jacs.5b08980), it removes the vast majority of AnI-sites: After digestion, the peptides containing clicked AnI-residues remain bound to the resin and are not analysed, which prevents a specificity evaluation based on AnI-modified peptides. Accordingly, we can't provide the suggested analysis of AnI-containing peptides.

We agree that enrichment strategies that preserve modified peptides are attractive and offer unique opportunities. For example, a recent publication demonstrates AnI-peptide-centric cell-selective proteomics elegantly combined with multiplexing (Schiapparelli, Lucio M. *et al.* (2022), doi: 10.1523/JNEUROSCI.0707-22.2022). Nonetheless, for reasons already touched upon by the reviewer, we believe that choosing a protein-level enrichment strategy has provided strong benefits for our particular study goals:

Given the low methionine frequency in the proteome, and the substoichiometric AnI incorporation (see Sup Fig 1), we expected very low AnI-protein amounts in small cell populations from tissues (e.g. around 10-20 % of all tumor cells as determined by FACS (Supplementary Fig. 5)). Our interest in extracellular proteins further increased the demands on assay sensitivity as secreted proteins typically miss the starting methionine due to signal peptide cleavage, and crucial intercellular signalling proteins such as cytokines are typically very small (for example, mature human EGF has only a single methionine) and often have particularly low abundance. Therefore, whole protein pulldowns that take all peptides of specifically enriched proteins into account significantly improve the reliable detection and precise quantification for protein classes that are in the focus of our study. Besides boosting sensitivity due to more available peptides per protein and consequently higher chances for proteotypic peptides that are MS accessible, our strategy also decreases single-peptide protein identifications.

To the best of our knowledge, the aforementioned study by Schiapparelli *et al.* is the most comprehensive published analysis of AnI-peptide-level enrichment to date. The proteome coverage ranges from 3521 identified AnI-peptides corresponding to 1932 proteins (including only 37 proteins with annotated extracellular location, according to Figure 1 and the extended data table) in HEK cells after 24 hours of *in vitro* labeling, to 3168 proteins from *in vivo* labeled cortical glutamatergic neurons. These results support our notion that a comprehensive analysis of intercellular signaling might still be very challenging with current AnI-peptide-level enrichment-based strategies.

We acknowledge the unique advantages of peptide-level enrichment methods and discuss differences to protein-level enrichments now in the discussion:

"Direct analyses of labeled peptides offer straightforward solutions for multiplexed cell type-resolved proteomics and the evaluation of enrichment specificity⁸⁷. However, with thorough background interference controls, analyses of all peptides from labeled proteins yields increased sensitivity and protein quantification accuracy."

We strongly agree that assessing enrichment specificity with well-designed controls is essential when analyses are not directly based on modified peptides. As described in more detail below, we tightly controlled all of our individual experiments following the same proven principles many other groups used previously (e.g. Alvarez-Castelao, Beatriz *et al.* (2017), doi:10.1038/nbt.4016,

Azizian, Nancy G. *et al.* (2021), doi: 10.1021/acs.jproteome.0c00666, Rayaprolu, Sruti *et al.* (2022), doi:10.1038/s41467-022-30623-x). We used wild-type (WT) (not MetRS*⁻-expressing) cells as negative controls to measure unspecific enrichment of proteins and evaluate their relative contribution to the signal in MetRS*⁺ samples. We considered only proteins with at least 3-fold enrichment compared to negative controls or exclusive identifications in MetRS*⁺ samples for further analysis. Our strategy thereby aligns well with the most stringent filtering strategies used in previous AnI enrichment studies (see, for example, Alvarez-Castelao *et al.* 2017).

How many times was the negative control repeated?

We thank the reviewer for bringing to our attention that our enrichment control procedure could benefit from a clearer description.

We used WT cell controls that were treated and processed alongside with MetRS*⁺ samples for each individual experiment to assess unspecific background and exclude unspecifically enriched proteins. We have now extended the description of our general specificity control strategy in the main text:

“...We applied the described filtering strategy to all subsequent MetRS*⁺ experiments in this study, using at least three experiments with wild-type cells as negative controls for corresponding MetRS*⁺ sample groups to define specifically enriched proteins and ensure high-confidence cell selectivity. ...”

We further extended the Methods section to explicitly define negative controls for each experiment, as follows:

“For technical experiments (Fig. 1 and 2), PDAC MetRS*⁺ tumor comparison experiments (Fig. 5), and serum secretomics experiments with the PDAC lines 8661 and 8513 (Fig. 6), corresponding WT controls were used in triplicates for each PDAC line. We used aggregated control sample groups for multiple experimental groups in the co-culture experiments (Fig. 3 and 4): Three BMM WT samples were used to control BMM MetRS*⁺ samples cultured in isolation. A group of four co-cultured BMM WT + PDAC WT samples (one with each of the four PDAC lines) was used as controls for all BMM MetRS*⁺ + PDAC WT co-culture samples. Both solo and co-cultured PDAC MetRS*⁺ samples were controlled with the more conservative corresponding co-culture control samples (PDAC WT + BMM WT in triplicates for each of the for PDAC lines).”

Our PRIDE upload contains every individual measured sample together with complete data tables and a sample description list that provides context on where each sample was used.

Moreover, for transparency and convenient access, we provided supplementary tables with the proteomics data of biological experiments and MetRS*⁻-to-Ctrl sample ratio columns that allow readers to quickly evaluate the amount of background interference for individual proteins and samples in each experiment.

Was it consistent each time or did the identities of proteins change and by how much, e.g. if you prepare 10 biological replicates of the negative control with a tight protein identification FDR what is the overlap, 50%, 60%, 70%, etc?

This is an interesting question. To address this point and a related point asked by reviewer 3, we have performed a series of additional experiments and analyses.

We have added a new figure (Supplementary Fig. 2) that provides insights on data completeness and CVs between replicates, using our enrichment and acquisition method benchmark datasets as an example.

Supplementary Fig. 2: Technical reproducibility of MetRS*-based cell-selective proteomics experiments, related to Fig. 1. **a)** Data completeness and **b)** precursor coefficients of variation (CVs) of MetRS* and Ctrl samples processed with different AnI enrichment workflows (see Fig. 1a). **c)** Data completeness, **d)** precursor CVs, and **e)** intensity ratios of proteins identified in both MetRS* and Ctrl samples after processing with our enrichment workflow and MS analysis by data-dependent acquisition (DDA) or data-independent acquisition (DIA).

With 84.4 % (DDA) and 98.7 % (DIA) of all identifications quantified in all three replicates (93 % (DDA) and 99.6 % (DIA) in at least two out of three), we achieved excellent data completeness in MetRS* samples using our workflow (new Supplementary Fig. 2a and c). Coefficients of variations (CVs) between our workflow replicates were lower than CVs of the DST method and comparable to the DBCO-agarose-based workflow (new Supplementary Fig. 2b), which, however, had a much higher unspecific background (see Fig. 1). Importantly, our direct comparison to conventional FACS-based cell-selective proteomics with tumor samples also showed lower CVs and therefore better technical reproducibility with our method (see Fig. 2).

Compared to MetRS* samples, negative controls had a lower data completeness and higher CVs. We attribute this primarily to the high signal-to-noise ratio in this experiment and the consequently very low peptide intensities in control samples, especially with our method. Higher technical variation is expected in samples with low signal intensity, due to the heteroscedastic nature of MS data (with variance inversely proportional to abundance).

As expected, data-independent acquisition (DIA) strongly increased identifications in both MetRS* samples and controls, which also increased the overlap of identified proteins from 909 to 4985 protein groups (new Supplementary Fig. 2e). Although DIA median MetRS*/Ctrl log₂ intensity ratios shifted to a lower level compared to DDA, a median of 8.6 (almost 400fold higher median intensity of overlapping proteins in MetRS* samples compared to negative controls) still indicated an extremely high signal-to-noise ratio.

Of note, after filtering for overlapping protein ratios between both acquisition methods (matched by gene names, due to differences in protein grouping), the ratio distribution shift between DDA and DIA data remained similar. This suggests that the shift is not driven by the increased identifications with DIA, but rather by acquisition method- and software-inherent differences in protein quantification.

To address the reviewer's question directly, we have performed additional experiments, which included 10 independent biochemical negative controls, as suggested.

Specifically, we prepared and measured three new replicates of MetRS* 8661 cells together with ten replicates of corresponding WT 8661 cells as controls (10 million cells per sample, labeling for 8 h in Met-free media with 4 mM Anl). These new data confirm our previous results and provide additional evidence for the technical reproducibility of our workflow. We include these results in a new Supplementary Figure 4 (see below):

Overall detected peptide intensities in negative controls were very low (new Supplementary Fig. 4a), providing strong evidence for the high enrichment specificity of our workflow. The log₂ intensity ratios between proteins identified in both MetRS* and WT control samples were very high (new Supplementary Fig. 4c), indicating minimal background signal interference in the vast majority of proteins quantified in MetRS* samples.

As a consequence of the high enrichment specificity and therefore low peptide intensities in WT controls, data completeness was again lower in control samples compared to MetRS* samples (new Supplementary Fig. 4b). Notably, results remained very consistent when control samples were divided into groups of three and used separately to evaluate signal-to-noise ratios (MetRS*/Ctrl log₂ intensity ratios) (new Supplementary Fig. 4c). The vast majority of proteins with sparse identifications in controls had high ratios far above the specificity cutoff (new Supplementary Fig. 4d). Conversely, the majority of proteins with lower MetRS*/Ctrl ratios had very high data completeness (new Supplementary Fig. 4d). Together, these results show not only that background protein interference was minimal, but also that three negative control samples were sufficient to reliably detect most unspecifically enriched proteins.

The reported protein numbers are high and thus the authors need to be rigorous in establishing those numbers are not simply due to non-specific interactions. Proving labeled peptides exist in their data for a statistically significant number of the proteins that were identified (e.g. of the 10K proteins) will go a long way to establish the legitimacy of the approach.

We thank the reviewer for bringing up this important point.

Although we are unable to provide information on enriched Anl-peptides for technical reasons explained above, we hope to have convinced through our rigorous enrichment control strategy and detailed analysis of technical reproducibility, as shown in our new Supplementary Figures, that unspecific binding is well controlled in our experiments.

Of note, our general control and filtering strategy based on quantitative comparisons to background controls is a well-established and widely used concept in MetRS*-based experiments and similar methods, such as proximity labeling-based cell-selective proteomics techniques, and can be considered a proven state-of-the-art (Alvarez-Castelao, Beatriz *et al.* (2017), doi:10.1038/nbt.4016, Azizian, Nancy G. *et al.* (2021), doi: 10.1021/acs.jproteome.0c00666, Rayaprolu, Sruti *et al.* (2022), doi:10.1038/s41467-022-30623-x, Alvarez-Castelao, Beatriz *et al.* (2019), doi: 10.1038/s41596-018-0106-6, Liu, Yan *et al.* (2017), doi: 10.1038/s41596-018-0106-6, Prabhakar, Priyadharshini *et al.* (2023), doi: 10.1002/glia.24304).

Apart from technical aspects, our experiments provide new biological insights into non-cell autonomous mechanisms of cancer and confirm observations from previous studies independent of MetRS*. For example, the pro-inflammatory secretory program of LPS treated macrophages detected in our study aligns well with previous studies from our and other labs (Meissner, Felix *et al.*, (2013), doi: 10.1126/science.1232578; Eichelbaum, Katrin *et al.* (2012), doi: 10.1038/nbt.2356). These studies are based on serum-free and azidohomoalanine labelling-based secretomics, respectively, confirming the consistency across technologies and the high cell-specificity of our data. Moreover, PDAC subtype-specific proteins detected in tumor-bearing mouse serum are consistent between different experiments in our study (e.g. Fig. 5a and Fig. 6f) and corroborate effective cell-selective protein enrichment, considering the extreme dominance and high dynamic range of serum proteins. (Geyer, Philipp E. *et al.* (2017), doi: 10.15252/msb.20156297).

We added one sentence to the discussion to acknowledge confirmatory results obtained by previous studies using distinct proteomics methods:

“Our findings not only recapitulated the pro-inflammatory secretory programs of macrophages as determined previously by distinct proteomics methods independent of MetRS*^{12,38} but identified for example 68 cancer cell-derived proteins with cytokine function in serum-containing culture media.”

Summary- overall the paper is potentially an important contribution to the growing area of cell specific proteomics. While I'm not an expert in pancreatic cancer it appears, some interesting discoveries were made using this approach. Cell specific proteomics is potentially an important companion to the emerging area of single cell proteomics (SCP). The potential increase in the number of proteins identified in these studies is about 10X beyond what the current capability is for single cell proteomic methods so it will allow the study of single cell types within complex tissue matrices as SCP matures. And cell specific BONCAT allows you to ask different types of questions.

We thank the reviewer for acknowledging the complementary nature of our work in the context of other emerging proteomic technologies.

Other questions.

Line 396-397: Is this over 8000 mice

We thank the reviewer for pointing out the ambiguity in this sentence. We have now rephrased it to clarify that the numbers are referring to the PDAC subtypes and not individual mice:

“Particularly, serum samples from mice bearing the 8661 (classical) and 8513 (mesenchymal) PDAC subtype tumors showed a good signal-to-noise ratio, ...”

Reviewer #2:

Swietlik et al. describe an improved method to selectively label and profile proteome contents in pancreatic cancer model systems. The system utilizes azidonorleucine to label and subsequently profile proteome using mass spectrometry. The study is well-designed and clearly written and demonstrates differential proteome coverage over the flow cytometry-based dissociation/proteome profiling methods, allowing for detection of some of the less abundant proteins, such as cytokines. The study also demonstrates significant differences in how distinct subclasses of pancreatic cancers, such as basal and classical, regulate myeloid cell polarization and composition of extracellular matrix. Overall, this work is very interesting, although some minor questions remain as to utility of this platform for diagnostic purposes, especially in the setting of human malignancy.

We thank the reviewer for the positive evaluation of our study.

Minor points:

- 1) A demonstration of how well this method can be applied in the human malignancy setting would be of great interest. At the very minimum, there needs to be a discussion of how the authors envision pipelining such protocols for diagnostic purposes.

We thank the reviewer for this comment. We have now added a new section to the discussion which addresses clinical translation:

“... Although our cell type-specific metabolic labelling approach cannot be directly applied to human cancer patients, it offers several new possibilities for clinical translation. MetRS* transduced human premalignant cells (e.g. from pancreatic intraepithelial neoplasia (PanIN) or intraductal papillary mucinous neoplasm (IPMNs)), as well as PDAC cells and organoids (e.g. from invasive tumors representing various stages of PDAC progression, differences in metastatic capacity or molecular subtypes), can be transplanted into immunodeficient mice or mice with a humanized immune system¹¹⁵. Subsequent MetRS*-based proteomic profiling of tumors and body fluids, such as the blood, enables not only a deeper understanding of PDAC development, progression and subtype specification, but has also potential of biomarker identification. So far, biomarkers for PanIN/IPMN and early PDAC detection, subtype classification, prognostic and therapeutic stratification, and the monitoring of targeted interventions are widely lacking¹¹⁶⁻¹¹⁸. MetRS* based proteomic profiling holds the promise of biomarker discovery in tumors and circulation, which can be subsequently tested and validated in prospective studies in cancer patients. ...”

- 2) ECM proteins are typically abundantly expressed in pancreatic cancer. How does the new method compare to other mass spec-based approaches (not based on flow cytometry) in identifying differences in ECM composition? Does the improved coverage in the new method add any information in this regard?

We thank the reviewer for this interesting question.

Recent pioneering work has demonstrated the high value of cancer cell-selective matrisome analyses in primary tumors and metastases (Di Martino, Julie S. *et al.* (2022), doi: 10.1038/s43018-021-00291-9, Tian, Chenxi *et al.* (2019), doi: 10.1073/pnas.1908626116, Tian, Chenxi *et al.* (2020), doi: 10.1158/0008-5472.can-19-2578). However, investigating extracellular proteins with cell type-resolution is difficult using conventional MS-based proteomics techniques. The previous studies relied on species differences between transplanted cancer cells and host stromal cells to distinguish the cell type origin of tumor matrix proteins in xenotransplantation models. While undoubtedly facilitating important discoveries, the approach has technical limitations: Both the need for species-distinguishing peptides and the potential for dynamic range issues caused by the co-analysis of abundant host proteins can reduce the cell-selective

matrisome coverage. Moreover, only transplanted cells can be studied and xenotransplantations require immunodeficient mice, which alters the tumor development and the matrix composition (Winkler, Juliane *et al.* (2020), doi: 10.1038/s41467-020-18794-x, Cox, Thomas R. (2021), doi:10.1038/s41568-020-00329-7, and see also e.g. Supplementary Fig. 7).

Accordingly, our approach offers several key advantages over traditional proteomics methods, including the ability to achieve deep cell-selective coverage of the matrisome for any desired cell type in fully immunocompetent mice.

We included these points now in the discussion:

“Here, we focused on another advantage of the technique - the accessibility of extracellular proteins for click chemistry enrichment. Anl labeling facilitates the cell-selective analysis of secreted proteins in tissue or body fluids, which is of great interest and difficult to achieve with conventional techniques. For example, recent pioneering work has demonstrated the high value of cancer cell-selective matrisome analyses in primary tumors and metastases^{73,95}, but relied on xenotransplants and immunocompromised mice to achieve cell-selectivity. In comparison, MetRS*-based cell-selective proteomics can provide additional value by overcoming the need for species-distinguishing peptides and avoiding potential dynamic range issues caused by the co-analysis of abundant host proteins, which can both reduce the cell-selective matrisome coverage. Moreover, in principle any cell type can be studied, and without the need for immunosuppression.”

Furthermore, we took the opportunity to compare our data to a PDAC matrisome study by Tian *et al.* (Tian, Chenxi *et al.* (2019), doi: 10.1073/pnas.1908626116). To the best of our knowledge, this study provides the most comprehensive published cell type-resolved PDAC matrisome analysis (based on xenotransplantations), and also one of the most comprehensive non-cell-selective PDAC matrisome proteomics datasets available.

As many ECM-focused proteomics studies, Tian *et al.* used a specialized sample preparation workflow to enrich ECM proteins from tissue. ECM enrichment methods usually use decellularization or sequential extraction of soluble proteins from tissue homogenates. This can achieve high ECM protein purity, and typically effectively captures ECM components with low solubility, including core matrix proteins like collagens (Krasny, Lukas *et al.* (2021), doi: 10.1080/14789450.2021.1984885). However, matrix-associated proteins are often underrepresented and better covered by ECM enrichment-independent methods (Krasny, Lukas *et al.* (2016), doi: 10.1042/bcj20160686, Krasny, Lukas *et al.* (2018), doi: 10.1016/j.jprot.2018.02.026), as demonstrated also by our data.

We have summarised the matrisome coverage for both their cell-selective data and their non-cell-selective mouse PDAC data in table 1.

	Xenograft high confidence (cell-selective)	Stromal-cell-derived	Cancer-cell-derived	Both-derived		Total matrisome coverage mouse PDAC (non-cell-selective)
Collagens	33	17	4	12		50
Glycoproteins	69	45	11	13		96
Proteoglycans	10	8	2	0		13
ECM regulators	61	43	8	10		85

	Xenograft high confidence (cell-selective)	Stromal-cell-derived	Cancer-cell-derived	Both-derived		Total matrisome coverage mouse PDAC (non-cell-selective)
ECM-affiliated	31	9	17	5		46
Secreted factors	16	7	7	2		31
Total	220	129	220	129	220	129

Table 1: Example data from non-cell-selective and xenograft-based cell-selective PDAC matrisome coverage (Tian, Chenxi et al. (2019), doi: 10.1073/pnas.1908626116)

	Total PDAC cancer-cell-derived	Mesenchymal PDAC cancer-cell-derived	Classical PDAC cancer-cell-derived
Collagens	30	29	21
Glycoproteins	107	97	92
Proteoglycans	14	15	13
ECM regulators	126	114	110
ECM-affiliated	55	53	50
Secreted factors	73	67	59
Total	405	405	405

Table 2: MetRS-based cell-selective PDAC matrisome coverage in our study.*

In total, we quantified over four times more cancer cell-derived matrisome proteins using fully immunocompetent mice. Compared to the non-cell-selective PDAC matrisome experiments, we identified less collagens but more matrix-associated proteins, particularly ECM regulators and secreted factors.

In conclusion, deep MetRS*-based cell-selective tissue proteomics opens up completely new perspectives for cell type of origin-resolved ECM analysis. It further demonstrates coverage advantages even compared to non-cell-selective approaches with dedicated ECM enrichment protocols, covering especially matrixome-associated proteins particularly well, which include many intercellular messengers such as cytokines. For even more comprehensive profiling of core matrix proteins, AnI enrichment could be combined with ECM-enrichment workflows in the future.

Reviewer #3:

In this manuscript, Swietlik et al. applied a combination of cell-selective metabolic labeling (MetRS-based Anl labeling) and MS-based proteomics to study pancreatic ductal adenocarcinoma (PDAC) subtypes and their tumoral microenvironment (TME). In a first step, they optimized the labeling and MS analysis workflow for a deep cell proteome and secretome coverage and demonstrated its superiority to cell sorting-based methods *in vivo*.

In a second part, they applied this optimized protocol to the analysis of PDAC cells from two different subtypes (classical and mesenchymal) co-cultured with macrophages. Using this cell-selective labeling, they were able to selectively study proteomes and secretomes of both PDAC cells and macrophages.

They applied a similar strategy to an *in vivo* study of cancer cell-derived matrix proteins through the orthotopic transplantation of Anl labeled classical or mesenchymal PDAC in mice. Finally, they studied PDAC subtype cell-derived proteins in circulation in the serum of orthotopically transplanted mice.

Although, the techniques used in this study are not new (use of CuAAC for metabolic labeling, combination of MetRS-based Anl labeling with MS proteomics, application of this method *in vivo*, DIA MS acquisition to improve protein identification), the authors demonstrated their combination to be highly efficient for a cell-selective deep proteome coverage. More innovatively, they applied these techniques to secretome analyses both in culture and *in vivo* which, to our knowledge, has never been described in the literature so far.

Considering the rising incidence and high lethality of pancreatic ductal adenocarcinoma and the need of a better comprehension of the tumor signaling with its microenvironment, the work presented in this manuscript is of high interest. In addition to provide new insights in the PDAC cell communication with the TME, it also reveals differences between classical and mesenchymal subtypes. Moreover, it demonstrates, through the analysis of serum from PDAC transplanted mice, that this strategy can be used to access cancer cell-derived proteins directly in body fluids which might be essential for the understanding of long-distance signaling in tumor progression and metastasis.

Although, on a statistical point of view, the findings described in this manuscript need to be confirmed by larger studies, this work opens the way to a better comprehension of PDAC subtypes signaling with its TME that may eventually lead to the identification of new biomarkers or therapeutic targets. It also demonstrates the feasibility and usefulness of cell-selective proteomics for short- and long-distance cell signaling studies that may be applied to other types of cancer. Therefore, I believe this manuscript is of great interest to the readers of Nature communications, provided that the authors can address the following points:

We thank the reviewer for the thorough and positive review and also the helpful suggestions.

- Figure 1c and 1e: although the standard deviations are shown on the figures, it only represents the variability in the total number of protein identifications for each method. It would be interesting to know how many proteins (or which percentage) overlap between the 3 replicates for each method to assess the technical variability. Moreover, coefficient of variation on the protein intensities over the 3 technical replicates should be provided to assess the efficiency of the methods for protein quantification.

We thank the reviewer for this comment. To address this point and related questions by reviewer 1, we now provide a new supplementary figure that includes additional analyses of the enrichment and acquisition method comparison experiments.

Supplementary Fig. 2: Technical reproducibility of MetRS*-based cell-selective proteomics experiments, related to Fig. 1. **a)** Data completeness and **b)** precursor coefficients of variation (CVs) of MetRS* and Ctrl samples processed with different AnI enrichment workflows (see Fig. 1a). **c)** Data completeness, **d)** precursor CVs, and **e)** intensity ratios of proteins identified in both MetRS* and Ctrl samples after processing with our enrichment workflow and MS analysis by data-dependent acquisition (DDA) or data-independent acquisition (DIA).

With 84.4 % (DDA) and 98.7 % (DIA) of all identifications quantified in all three replicates (93 % (DDA) and 99.6 % (DIA) in at least two out of three), we achieved excellent data completeness in MetRS* samples using our workflow (new Supplementary Fig. 2a and c). Coefficients of variations (CVs) between our workflow replicates were lower than CVs of the DST method and comparable to the DBCO-agarose-based workflow (new Supplementary Fig. 2b), which, however, had a much higher unspecific background (see Fig. 1). Importantly, our direct comparison to conventional FACS-based cell-selective proteomics with tumor samples also showed lower CVs and therefore better technical reproducibility with our method (see Fig. 2).

Compared to MetRS* samples, negative controls had a lower data completeness and higher CVs. We attribute this primarily to the high signal-to-noise ratio in this experiment and the consequently very low peptide intensities in control samples, especially with our method. Higher technical variation is expected in samples with low signal intensity, due to the heteroscedastic nature of MS data (with variance inversely proportional to abundance).

As expected, data-independent acquisition (DIA) strongly increased identifications in both MetRS* samples and controls, which also increased the overlap of identified proteins from 909 to 4985 protein groups (new Supplementary Fig. 2e). Although DIA median MetRS*/Ctrl log₂ intensity ratios shifted to a lower level compared to DDA, a median of 8.6 (almost 400fold higher median intensity of overlapping proteins in MetRS* samples compared to negative controls) still indicated an extremely high signal-to-noise ratio.

Of note, after filtering for overlapping protein ratios between both acquisition methods (matched by gene names, due to differences in protein grouping), the ratio distribution shift between DDA and DIA data remained similar. This suggests that the shift is not driven by the increased identifications with DIA, but rather by acquisition method- and software-inherent differences in protein quantification.

Protein group CVs of the enrichment and acquisition methods compared in Figure 1.

We included precursor CVs in the new Supplementary Figure 2 for consistency with our FACS and MetRS* comparison (Fig. 2d). Protein CVs follow the same trends: CVs in MetRS* samples were very low, while CVs between Ctrl samples were higher, except for the DBCO method, which showed a comparably high degree of unspecific enrichment and therefore a much stronger signal in Ctrl samples (see Fig. 1b - c). As expected, DIA drastically improved identifications, data completeness, and CVs, especially in the low signal intensity Ctrl samples. Overall, DIA analysis led to robust results, even in negative Ctrl samples of experiments with very high AnI-protein yields and highly specific enrichment, and consequently extremely low background signal (see also new Supplementary Fig. 4).

- Figure 1g: Considering that a large proportion of signaling proteins have a low molecular weight (e.g., cytokines) and that quantification of proteins on single peptides are usually less accurate. It

would be interesting to mention, on the figure or in the text, the proportion of proteins identified with only 1 peptide or with 2 peptides and more.

This is an important point. We do indeed see elevated numbers of proteins identified with a single peptide in this fraction of the proteome. Still, the majority of identified signalling proteins were identified with two or more peptides. As suggested, we have now added this information to the description of Fig 1g:

“[...]Of those, 103 protein groups are known ligands for intercellular communication according to CellPhoneDB39, including 46 with described cytokine function (Fig. 1g).] **Despite their often small size and low abundance, 83 (81 %) and 41 (89 %) of the detected intercellular signalling proteins and cytokines were identified with at least two peptides.**”

- Line 199: The authors claimed they doubled the number of protein identifications with their method in comparison to reference #28. However, this comparison is biased by the fact that they used two different generation of MS instruments (Exploris 480 for this study; Fusion Lumos for ref #28). The authors should remove this statement or discuss the influence of the MS type on the results.

Having worked with MS instruments from both generations, we are confident that machine type performance differences are much smaller than the difference in coverage between both studies. However, we agree that it does bias the comparison. As we do not have access to a Lumos to evaluate the instrument-related proportion of the coverage gains, we agree to remove the statement.

- Line 202-204 and Fig 2e: While it is understandable that flow cytometry method cannot capture cell released proteins, the authors should comment on the reasons why the cell-selective labeling is less efficient in capturing transmembrane proteins.

This is a very interesting question. We attribute this primarily to differences in the sample preparation, especially the lysis buffer composition:

Sorted cells were lysed in a sodium deoxycholate (SDC)-based buffer, a strong ionic detergent which supports efficient transmembrane protein extraction and digestion (Varnavides, Gina *et al.* (2022), doi: 10.1021/acs.jproteome.2c00265, Alfonso-Garrido, Javier *et al.* (2015), doi: 10.1007/s00216-015-8732-0). Moreover, lysates were not cleared, which has been shown to further improve membrane protein coverage in SDC-based protocols (Kulak, Nils A. *et al.* (2014), doi: 10.1038/nmeth.2834).

Most commonly used detergents that are very effective for membrane protein extraction are detrimental for the reaction kinetics of copper-catalyzed click chemistry (Yang, Yinliang *et al.* (2013), doi: 10.3390/molecules181012599). Instead, we used a phosphate-free high salt buffer with high initial guanidine concentrations and the milder zwitterionic detergent CHAPS for AnI-protein enrichment, we cleared lysates to reduce unspecific background binding to beads, and we strongly diluted chaotrope and detergent concentrations before starting the click chemistry capture reaction. While these steps were crucial for reaching superior click chemistry yields together with good overall tissue protein extraction, the protocol is likely less effective for membrane protein recovery than strong detergent-based lysis workflows.

We have now added a comment on the transmembrane protein enrichment in samples processed with the FACS workflow:

“Flow cytometry-sorted samples showed, for example, enrichment of transmembrane proteins, likely facilitated by strong ionic detergent-based lysis, which enhances transmembrane protein extraction and digestion^{41,42} but can interfere with CuAAC reactions³⁴.”

- Supp Fig 4: For an easier reading of the figure, it would be appreciated to have the mention "classical" or "mesenchymal" next to the number of each cell line.

We thank the reviewer for this suggestion. We have adjusted the figure accordingly (and also the Supplementary Fig. 7).

- Supp Fig 4: This figure would deserve a better description in the text. While it is obvious that the PDAC cell proteomes are modified when co-cultured with BMM, it is not as clear for the secretome. Conversely, both BMM proteomes and secretomes are changing upon co-culture with PDAC cells.

We thank the reviewer this suggestion. We have clarified this point by adjusting the description of Supplementary Fig. 4 as follows:

“Principal component analyses (PCAs) showed reciprocal adaptations of cancer cells and BMMs to co-culture with changes of both global proteome expression and protein secretion, although less clear for PDAC secretomes (Supplementary Fig. 6). PCAs further indicated distinct differences between PDAC subtypes, and PDAC line-specific BMM responses.”

- Figure 3b: It is not clear in the legend if this figure represents GO terms enrichment for proteome data only or for proteome + secretome. By reading the text, it seems to be proteome only but it should be added in the legend. In this case, one can wonder why no GO enrichment analysis has been performed on secretome data.

This is correct, we initially only analysed GO term enrichment between global proteomes. However, we agree that a GO analysis of the secretomes may be interesting. We have now performed this new analysis, added it to Figure 3 and describe the results in the manuscript. This enrichment analysis does not only support our previous observations but indeed provides new insights:

New GO enrichment plots from new Fig. 3.

For example, increased detection of MHC I and other surface exposed transmembrane proteins in PDAC secretomes upon co-culture suggest increased shedding activity. Moreover, in addition to immunomodulatory signals and growth factors, enriched GO terms in co-cultured BMM

secretomes reflect, for example, the upregulation of matrix modulating enzyme secretion described in supplementary figure 5.

- Supp Fig 5: The proteins having significant abundance differences should be indicated on the figure (e.g., the 68 cytokine function proteins mentioned line 242).

We thank the reviewer for bringing this to our attention. The data displayed in Supplementary Figure 5 was already filtered to display only proteins with significantly differential abundance (ANOVA, FDR = 0.05 and S0 = 0.1), however, we forgot to indicate this in the figure legend. We have now added the selection criteria in the description.

- Line 268: Define the abbreviations LPS and TLR4

We thank the reviewer for pointing out the missing descriptions - we have added them now.

- Fig 3d: On a graphical point of view, the distinction between the 3 yellow bars for the 3 BMM cultures is not clear. In addition, the results do not show a clear relation between the PDAC subtype differential expression of signaling proteins and the expression of their receptors on the macrophage (moreover, it is not even commented on in the text). Therefore, there is a low interest in representing the data in that way and the figure should better focus on the differences between classical and mesenchymal subtypes.

We thank the reviewer for making us aware that this figure benefits from graphical improvements and a better description. We have now significantly revised the plots to resolve issues with the data visualization and to add additional information:

- We noticed that the dynamic range of the color gradient of receptor expression levels in our original version was not ideal. We have now introduced a three-color gradient that illustrates receptor expression differences and dynamics clearer.
- We have added statistical information to highlight significant differences at the ligand and receptor levels
- We have reorganised the figure to make space for larger text labels that are easier to read
- The three yellow bars have been enlarged for better differentiation

In addition, we have now highlighted an example of increased receptor expression on BMMs in PDAC co-culture with implications for macrophage polarization:

"The majority of detected PDAC signal corresponding receptors on BMMs showed stable expression, but some were regulated upon co-culture with cancer cells (Fig. 4a). Notably, BMMs upregulated Pvr (Poliovirus receptor) expression upon interaction with both PDAC subtypes. Pvr activation on macrophages has been linked to an anti-inflammatory phenotype⁷¹ and targeting the Pvr-Tigit axis is being explored as a potential cancer immunotherapy strategy⁷²."

- Line 285-287: Please reformulate this sentence which is unclear.

We have added statistical information to the figure and rephrased the sentence:

"Some proteins were secreted without significant differences between subtypes, such as Tgfb1, a known M2 promoter⁵⁷, or, Tnf (at much lower abundance, see also Supplementary Table 1), an important M2-suppressing factor in cancer⁵⁸."

- Line 296: There is no data (in fig 3d or even in table 1) to support the statement that PDAC subtypes expressed "high levels" of Cd47. The fig 3d only shows it is expressed at similar

levels between the two subtypes and table 1 do not show particularly high quantitative values for this protein.

Although we detected CD47 at above average abundance compared to the expression of other receptors and plasma membrane proteins (for example, it ranks 476 out of 1246 proteins with the GOCC annotation “plasma membrane” in 8661), and at higher abundance than in BMMs (for example, 2.5-4fold compared to the median intensity in isolated BMM proteomes), we agree that this detail might not be very meaningful biologically, especially without providing reference levels of other cell types. Therefore, we removed the statement.

- Line 333: Since, the q-value associated to “MHC class I protein complex” term enrichment in fig. 3b is not significant, it cannot be stated that the MHCI antigen presentation-related protein expression is elevated in mesenchymal PDAC cells in co-cultures.

We thank the reviewer for this correction. We deleted “MHCI” and now only refer to “antigen presentation-related protein expression”, which is justified as the term “antigen processing and presentation of peptide antigen” is significantly enriched in mesenchymal PDAC co-culture experiments.

- Fig 4d: GO terms above the figures could be presented as a table or as a figure of enrichment analysis rather than a simple list.

We agree and have rearranged Fig. 5 (previously Fig. 4) and present enriched GO terms now in table format.

- Line 413: the word “systematically” should be removed since Lamc1 and Lamb1 do not have higher abundance in Fig 5f.

We agree and have removed “systematically” from our description.

- Line 494-496: The statement should be moderate on the fact that only one mesenchymal PDAC cell line has been used for the analysis of serum circulating proteins, the second having been left aside based on the PCA in Supp fig 7.

We were referring to proteins detected in primary tumors in this section of the discussion - here, all four cancer cell lines were analysed and consistently showed these trends (see figure 6, formerly figure 5).

We have rephrased the discussion section to clarify which datasets we are referring to and narrowed conclusions to the selection of cancer lines tested in our study:

“Previous research has shown that mesenchymal PDAC tumors have a higher cellularity, less activated CAFs and a less pronounced desmoplastic reaction⁹⁷. Our cell-selective tumor analysis revealed that, among the lines we tested, mesenchymal cancer cells themselves produce significantly higher levels of matrisome proteins, particularly core matrix proteins, compared to classical PDAC cells. Furthermore, mesenchymal cancer cells exhibited a distinct matrisome signature that promotes epithelial-to-mesenchymal transition (EMT).”

- In general, statistical considerations should be commented on in the discussion: the use of only 3 replicate mice for in vivo experiments, the use of only 2 cell lines per PDAC subtypes, the use of one cell line per subtype in the mouse serum experiment. Although this can be explained by the amount of work it involved, these choices should be explained and the results commented on from a statistical point of view.

We agree and have added moderating statements to conclusions about PDAC subtypes in the discussion.

Furthermore, we now include an additional passage to make the reader aware of statistical limitations arising from the n-numbers:

“Our study has identified novel differentiating features among PDAC subtypes with high consistency in our selected models. However, the limited sample size, with only two cell lines per subtype and one line per subtype in the serum secretomics experiment, and the small number of replicates in *in vivo* mouse experiments (three mice per line) do pose limitations to our findings. Despite this, our results demonstrate the unique strengths of cell-selective proteomics analyses in uncovering disease mechanisms and provide a foundation for further research with larger sample sizes to statistically validate and expand upon these findings.”

REVIEWERS' COMMENTS

Reviewer #1 (Remarks to the Author):

The authors addressed my questions adequately and the paper in my view is good to go. The additional experiments added to the paper strengthened the paper.

Reviewer #2 (Remarks to the Author):

My comments have been addressed

Reviewer #3 (Remarks to the Author):

The authors have provided detailed answers to my questions and improved their manuscript by adding new comments. They also provide an additional supplementary figure demonstrating the very good technical reproducibility of the method. The figures on functional analysis and intercellular signalling have been improved for more clarity and thus provide new insights on the results of the study.

Therefore, I do recommend the publication of this work in Nature communications.

RESPONSE TO REVIEWERS' COMMENTS

Reviewer #1 (Remarks to the Author):

The authors addressed my questions adequately and the paper in my view is good to go. The additional experiments added to the paper strengthened the paper.

Reviewer #2 (Remarks to the Author):

My comments have been addressed

Reviewer #3 (Remarks to the Author):

The authors have provided detailed answers to my questions and improved their manuscript by adding new comments. They also provide an additional supplementary figure demonstrating the very good technical reproducibility of the method. The figures on functional analysis and intercellular signalling have been improved for more clarity and thus provide new insights on the results of the study. Therefore, I do recommend the publication of this work in Nature communications.

We are glad, all three reviewers kindly acknowledge that we have addressed all points satisfactorily and recommend the publication of our manuscript.